# Land cover change influence on atmospheric organic gases, aerosols, and radiative effects

Ryan Vella[1,2], Matthew Forrest[3], Andrea Pozzer[1,4], Alexandra P. Tsimpidi[5], Thomas Hickler[3,6], Jos Lelieveld[1,4], and Holger Tost[2]

[1]Atmospheric Chemistry Department, Max Planck Institute for Chemistry, Mainz, Germany
[2]Institute for Atmospheric Physics, Johannes Gutenberg University Mainz, Mainz, Germany
[3]Senckenberg Biodiversity and Climate Research Centre (SBiK-F), Frankfurt am Main, Germany
[4]Climate and Atmosphere Research Center, The Cyprus Institute, Nicosia, Cyprus
[5]Institute for Energy and Climate Research, IEK-8: Troposphere, Forschungszentrum Jülich GmbH, Jülich, Germany
[6]Department of Physical Geography, Goethe University, Frankfurt am Main, Germany

**Correspondence:** Ryan Vella (ryan.vella@mpic.de)

**Abstract.**

Biogenic volatile organic compounds (BVOC) are emitted in large quantities from the terrestrial biosphere and play a significant role in atmospheric gaseous and aerosol composition. Secondary organic aerosols (SOA) from BVOC oxidation affect the radiation budget both directly through scattering and absorption of sunlight and indirectly through modifying cloud properties.

Human activities have extensively altered natural vegetation cover, primarily by converting forests into agricultural land. In this work, a global atmospheric chemistry-climate model coupled with a dynamic global vegetation model was employed to study the impacts of perturbing the biosphere through human land use change, consequently exploring changes in BVOC emissions and atmospheric aerosol burden. A land use scheme was implemented to constrain the tree plant functional type (PFT) cover based on land transformation fraction maps from the year 2015. Two scenarios are evaluated: (1) comparing present-day land

cover, which includes areas deforested for crops and grazing land, with the potential natural vegetation (PNV) cover simulated by the model, and (2) an extreme reforestation scenario where present-day grazing land is restored to natural vegetation. We find that, compared to the PNV scenario, present-day deforestation results in a 26% reduction in BVOC emissions, which decreases the global biogenic SOA (bSOA) burden by 0.16 Tg (a decrease of 29%), while the total organic aerosol (OA) burden decreases by 0.17 Tg (a reduction of 9%). On the other hand, the extreme reforestation scenario, compared to present-day land

cover, suggests an increase in BVOC emissions by 22%, which increases the bSOA by 0.11 Tg and total OA burden by 0.12 Tg, an increase of 26% and 6%, respectively. In the present-day deforestation scenario, we estimate a positive total radiative effect (aerosol + cloud) of 60.4 mW m$^{-2}$ (warming) compared with the natural vegetation scenario, while in the extreme reforestation scenario, we report a negative effect (cooling) of 38.2 mW m$^{-2}$ compared to the current vegetation cover.

# 1 Introduction

Human activities have significantly modified the natural vegetation cover, primarily through the conversion of forests into agricultural land. It is estimated that approximately half of the Earth's land surface has been affected by human activities (Hurtt et al., 2011). Land cover change (LCC) has a substantial impact on the Earth system as the biosphere plays a central role in major biophysical and biogeochemical cycles, as well as feedbacks with the atmosphere (Bonan, 2008). Forests store 45% of terrestrial carbon and can sequester large amounts of carbon (Field and Raupach, 2004). Furthermore, forests play a significant role in sustaining the hydrological cycle through evapotranspiration, which influences cloud formation, the onset of precipitation, and consequently surface temperatures (Betts et al., 2004; Vicente-Serrano et al., 2015). Land cover plays a crucial role in surface albedo, with dense forests capable of absorbing up to 90% of solar radiation (Forster et al., 2007). In high latitudes, forests can mask the high albedo of snow, leading to planetary warming through increased solar heating of the land (Bonan, 2008).

The terrestrial biosphere is also the primary source of biogenic volatile organic compounds (BVOCs) emissions such as isoprene and various terpenes, accounting for around 90% of the total VOC emissions to the atmosphere (Guenther et al., 1995). Reported global isoprene emissions, the primary BVOC, span 412 to 682 Tg yr$^{-1}$ (e.g., Sindelarova et al., 2014; Guenther et al., 2006; Vella et al., 2023a). BVOCs are highly reactive and short-lived, with lifetimes typically ranging from minutes to hours. Upon emission, they rapidly interact with tropospheric oxidant gases, thereby exerting a substantial influence on the oxidation capacity of the atmosphere (Lelieveld et al., 2008; Atkinson, 2000; Atkinson and Arey, 2003).

The short lifespan of BVOCs stems from their rapid oxidation upon release from the canopy. This oxidation primarily involves the OH radical as well as other key oxidizing agents like $O_3$ and $NO_3$ radicals (Shrivastava et al., 2017). These reactions produce various lower volatility oxidation products that tend to partition into the aerosol phase, leading to the formation of biogenic secondary organic aerosols (bSOA) (Kavouras et al., 1998; Spracklen et al., 2011).

Isoprene-derived oxidation products mainly contribute to the condensation onto existing aerosols (aerosol growth), whereas the oxidation products from monoterpenes, despite being less abundant compared to isoprene, play a crucial role in generating new particles (nucleation) (Jokinen et al., 2015). As a result, monoterpene precursors may have distinct climate impacts given their influence on the aerosol numbers. Through condensational growth, bSOA participates in the absorption and scattering of solar short-wave radiation, contributing to aerosol-radiation interactions (ARI). Furthermore, newly formed bSOA particles can grow into sufficient sizes to activate as cloud droplets, thereby modifying cloud properties such as albedo and lifetime, and effectively contributing to aerosol-cloud interactions (ACI) (Forster et al., 2007) In this study, we do not include organic new particle formation (NPF) and focus only on the role of organic precursors in supporting aerosol condensational growth.

BVOC emission rates are inherently linked to land cover, and LCC can ultimately affect the climate system by influencing short-lived climate forcers like aerosols (Scott et al., 2014). In this study, we investigate the changes in BVOC emissions following crop and grazing land expansion on potential natural vegetation (PNV). PNV refers to the type of vegetation that would naturally occur in a specific area under certain climate, soil, and environmental conditions without human influence. We use the chemistry-climate model EMAC coupled with the dynamic global vegetation model (DGVM) LPJ-GUESS. The

coupling is "one-way", whereby vegetation information in LPJ-GUESS is driven by climate states from EMAC, but feedbacks from vegetation to the atmosphere are suppressed, except for changes in BVOC emission rates. Thus, we focus on quantifying the aerosol burden and radiative effects driven purely by BVOC emissions from LCC, without accounting for changes in surface albedo, roughness length, or the hydrological cycle. Land cover change is simulated through a deforestation routine in LPJ-GUESS (which simulates PNV), systematically clearing crop and grazing land areas based on 2015 land cover data.

## 2 Model description and methods

### 2.1 The EMAC modelling system

The EMAC (ECHAM/MESSy Atmospheric Chemistry) model is a numerical chemistry and climate modelling system that contains submodels that represent tropospheric and middle atmospheric processes, as well as their interactions with oceans, land, and human activities. It originally combined the ECHAM atmospheric general circulation model (GCM) (Roeckner et al., 2006) with the Modular Earth Submodel System (MESSy) (Jöckel et al., 2005) framework and philosophy, modularising physical processes as well as most of the infrastructure into submodels that can be further developed to improve existing process representations. New submodels can also be added to represent new or alternative process representations.

Aerosols are treated using the submodel GMXe (Pringle et al., 2010), where aerosol microphysics are characterised by seven interactive lognormal modes that span the typical size range of aerosol species. These modes are further categorised into four hydrophilic (nucleation, Aitken, accumulation, and coarse) and three hydrophobic (Aitken, accumulation, and coarse) aerosol modes. The representation of all aerosols assumes spherical particles. The properties of aerosols in each mode are fully determined by the total mass (internal mixture of contributing species), density, number concentration, median radius, and width of the lognormal distribution. After each simulation step, aerosols may transfer between modes depending on size changes.

Organic aerosol species are additionally described by the Organic Aerosol Composition and Evolution (ORACLE) submodel (Tsimpidi et al., 2014), taking into account the partitioning between aerosols and the gas phase based on the volatility basis set (VBS) framework (Donahue et al., 2006). ORACLE describes the following organic aerosols (OA): Secondary organic aerosols from the oxidation of anthropogenic (aSOA) and biogenic (bSOA) VOCs; primary organic aerosols from emissions from fossil fuel and bio-fuel combustion (fPOA) and biomass burning (bbPOA); and secondary organic aerosols from their subsequent photochemical oxidation (fSOA and bbSOA, respectively) (Tsimpidi et al., 2016, 2017). ORACLE treats SOA mass yields for different lumped VOCs at varying saturation concentrations ($C^*$) in µg m$^{-3}$ at 298 K based on an assumed particle density of 1.5 g cm$^{-3}$. Isoprene exhibits mass yields across the range of $C^*$, with a peak of 0.03 at 10 µg m$^{-3}$ before declining sharply to 0.015 at 100 µg m$^{-3}$ and reaching zero at 1000 µg m$^{-3}$. Monoterpenes show significantly higher yields, starting at 0.107 for low $C^*$ (1 µg m$^{-3}$) and peaking at 0.600 for higher $C^*$ (1000 µg m$^{-3}$), highlighting the much greater contribution of monoterpenes to SOA formation, especially under conditions of higher saturation concentrations. More details on SOA mass yields in ORACLE can be found in Tsimpidi et al. (2014).

ORACLE employs a simple photochemical ageing scheme that effectively models the combined impacts of fragmentation and functionalisation of organic compounds. The module not only predicts the mass concentration of organic aerosol (OA) components but also predicts their oxidation state (expressed as O : C), enabling their categorisation into primary OA (POA, chemically unprocessed), freshly formed secondary OA (SOA, with low oxygen content), and aged SOA (highly oxygenated). By explicitly simulating the chemical conversion of OA from initial emissions to a highly oxygenated state during photochemical ageing, ORACLE facilitates tracking changes in OA hygroscopicity resulting from these reactions. This allows the computation of OA particle capability to serve as cloud condensation nuclei. The output from the ORACLE model, based on the described setup, has been compared with observational data in tropical regions (Hewitt et al., 2010; de Sá et al., 2019; Chen et al., 2009; Schmale et al., 2013; Tiitta et al., 2014; Zhang et al., 2010). We found that ORACLE provides surface OA concentrations within 60% of the observed values over these tropical forest regions. A thorough evaluation of ORACLE against aerosol mass spectrometer (AMS) measurements is provided in Tsimpidi et al. (2016). Details on the implementation of ORACLE (v2.0) in EMAC, along with a comprehensive model evaluation, can be found in Tsimpidi et al. (2018).

GMXe treats new particle formation (NPF) based on temperature, relative humidity, and sulfuric acid ($H_2SO_4$) concentration (Vehkamäki et al., 2002). In this setup, organics do not contribute to NPF but only participate in condensation through the VBS framework in ORACLE, as described above, where volatilities are governed by the oxidation of organic precursors.

Heterogeneous and gas-phase chemistry are treated through the MECCA submodel (Sander et al., 2019), employing the Mainz Isoprene Mechanism (MIM1) as the chemical mechanism (Pöschl et al., 2000; Jöckel et al., 2006), which includes over 100 gas-phase species and more than 250 reactions. MIM1 includes the following BVOC oxidation pathways: isoprene + OH, isoprene + $O_3$, and monoterpene oxidation (lumped species) with OH, $O_3$, $NO_3$, and $O(^1D)$. Dry deposition, sedimentation, and wet deposition processes are simulated using the submodules DDEP, SEDI (both Kerkweg et al., 2006) , and SCAV (Tost et al., 2006a), respectively.

Convective cloud processes are taken into account based on the approach proposed by Tost et al. (2006b), utilising the convection schemes from Tiedtke (1989) and Nordeng (1994). Convective cloud microphysics is solely based on temperature and moisture profiles, without accounting for the influence of aerosols on liquid droplet or ice formation processes. The vertical velocity distribution used for aerosol activation by grid-scale clouds in EMAC is calculated as the sum of the grid mean vertical velocity and the turbulent contribution, as detailed by Brinkop and Roeckner (1995). Large-scale stratiform clouds are described by the CLOUD submodel, which in the applied configuration, incorporates a two-moment cloud microphysics scheme for cloud droplets and ice crystals, as detailed by Lohmann et al. (1999, 2007); Lohmann and Ferrachat (2010) and Lohmann and Kärcher (2002). CLOUD solves prognostic equations for specific humidity, liquid cloud mixing ratio, ice cloud mixing ratio, cloud droplet number concentration (CDNC), and ice crystal number concentration (ICNC).

CLOUD incorporates a prognostic cloud droplet nucleation process to represent aerosol-cloud interactions in large-scale clouds (excluding convective warm clouds) using aerosol information provided by GMXe. This prognostic nucleation scheme is based on the aerosol activation parameterisation called the "unified dust activation framework" (UAF). UAF is based on Nenes and Seinfeld (2003) cloud activation parameterisation and includes dust-pollutant interactions during the activation process by taking into account the hydrophilicity of the dust using an adsorption theory together with the acquired hygroscopicity using

Köhler theory. This routine provides values for the hygroscopicity parameter ($\kappa$) and the cloud condensation nuclei (CCN) number concentration at 0.2% and 0.4% supersaturation for each aerosol size mode (Pringle et al., 2010).

For the evaluation of the radiative effects from clouds and aerosols, we employ the methods from Ghan (2013). The calculation of the net radiative effect from aerosol-radiation interactions (RE$_{\mathrm{ari}}$) involves determining the difference between the net top-of-the-atmosphere shortwave radiative flux ($F$) and the radiative flux, excluding the scattering and absorption of solar radiation by the aerosols ($F_{\mathrm{clean}}$). $F_{\mathrm{clean}}$ is computed in a separate radiation call within the radiation submodel RAD. Similarly, the radiative effect from aerosol-cloud interactions (RE$_{\mathrm{aci}}$) is derived by assessing the difference between $F_{\mathrm{clean}}$ and the flux that disregards the scattering and absorption caused by both clouds and aerosols ($F_{\mathrm{clear\text{-}sky, clean}}$).

EMAC includes land surface and vegetation models, enabling comprehensive studies of vegetation-atmosphere interactions. The Lund-Potsdam-Jena General Ecosystem Simulator (LPJ-GUESS) was the first vegetation model integrated into EMAC (Forrest et al., 2020; Vella et al., 2023a), and more recently, the Jena Scheme for Biosphere–Atmosphere Coupling in Hamburg (JSBACH) has also been incorporated as a submodel (Martin et al., 2024), further expanding EMAC's capacity to simulate land surface processes. In this work, we rely exclusively on LPJ-GUESS for vegetation calculations.

**LPJ-GUESS**

LPJ-GUESS (Smith et al., 2001, 2014) is a dynamic global vegetation model (DGVM) that features an individual-based approach to modeling vegetation dynamics. These dynamics are simulated as the emergent outcome of plant growth and competition for light, space, and soil resources among woody plant individuals and a herbaceous understorey in each of a number (50 in this study) of replicate patches representing random samples of each simulated locality or grid cell. The simulated plants are classified into 12 plant functional types (PFTs) discriminated by growth form, phenology, photosynthetic pathway (C3 or C4), bioclimatic limits for establishment and survival and, for woody PFTs, allometry and life history strategy. The LPJ-GUESS version used in this study (v4.0) currently provides information on potential natural vegetation (PNV), and it does not incorporate LCC. In this work, however, a custom deforestation routine was integrated to constrain the PNV using deforestation maps. The deforestation maps consist of values from 0 to 1, where a value of 1 signifies complete deforestation within the respective grid cell. The routine eliminates the tree PFTs after every simulated year and inhibits trees from establishing in the specified areas. This implementation allows us to constrain the vegetation cover and address the research questions presented in this work. However, the latest version of LPJ-GUESS (v4.1) features a more advanced land cover scheme, which will be incorporated into our current LPJ-GUESS version in future developments.

### 2.2 EMAC/LPJ-GUESS configuration

In this work, we use the standard EMAC/LPJ-GUESS coupled configuration, where the vegetation in LPJ-GUESS is entirely determined by the EMAC atmospheric state, soil type, N deposition, and $CO_2$ fluxes (Forrest et al., 2020), but there is no feedback from the vegetation to climate variables, except for terrestrial BVOC emissions. The roughness length and albedo are kept as constant background values. Albedo is derived from satellite climatologies, while the roughness length is based on

subgrid-scale orography and satellite-derived vegetation climatology. Vegetation changes do not feed back to the hydrological cycle. We use the native bucket model in ECHAM5, which employs fixed climatological vegetation (Hagemann, 2002). In this setup, BVOCs are interactive tracers that can be oxidized to form secondary organic aerosols. This means that BVOCs can influence the oxidant chemistry of the atmosphere; however, we do not quantify such impacts in this work, focusing solely on aerosol changes.

After each simulation day, EMAC computes the average daily values of 2-meter temperature, net downwards shortwave radiation, and total precipitation and passes these state variables to LPJ-GUESS. Vegetation information (leaf area index, foliar density, leaf area density distribution, and PFT fractional coverage) from LPJ-GUESS is then fed back to EMAC for the calculation of BVOC emission fluxes using EMAC's BVOC submodules. In this study, the BVOC fluxes in EMAC are calculated using the Model of Emissions of Gases and Aerosols from Nature (MEGAN) version 2.04 (Guenther et al., 2006).

MEGAN is based on the work of Guenther et al. (1993, 1995), where the BVOC emission flux is calculated as a function of PFT-specific emission factors, and non-dimensional activity factors. These activity factors consider sensitivities to the canopy environment, including parameters such as leaf area index (LAI), temperature, light, and leaf age. Notably, the current setup does not incorporate sensitivity to soil moisture. The parameterised canopy environment emission activity (PCEEA) algorithm is used, rather than the alternative detailed canopy environment model that calculates light and temperature at each canopy depth. The PCEEA algorithm calculates the light sensitivity within the canopy as a function of the daily average above-canopy photosynthetic photon flux density (PPFD), the solar angle and a non-dimensional factor describing the PPFD transmission through the canopy.

This setup employs BVOC-aerosol-vegetation feedbacks, making vegetation and BVOC emissions sensitive to changes in temperature and above-canopy radiation (excluding diffused radiation) resulting from aerosol interactions, nevertheless, these feedbacks are minimised by nudging meteorological fields toward observations. BVOC emissions from this model setup were evaluated and applied in other studies (e.g. Vella et al., 2023a, b).

## 2.3 Experimental design

The land cover scenarios were derived from the History database of the Global Environment (HYDE v3.2) (Klein Goldewijk et al., 2017). HYDE provides a wide range of land use products, encompassing both historical and projected data. To ensure an accurate representation, we rely on HYDE's "cropland" and "grazing land" products from 2015, derived from high-resolution satellite data. These products were transformed into deforestation fraction maps (Fig. 1) to constrain the vegetation in the model. Three experiments were conducted to assess the impact of human-induced LCC on the natural land biosphere and atmospheric composition. The initial model run used simulated PNV without any deforestation. Additionally, two more model runs were conducted, incorporating deforestation. The first scenario is aimed to represent present-day land cover employed by deforestation based on cropland and grazing land. This scenario is referred to as "Deforested Crop and Grazing Land" (DCGL) - Fig. 1a. DCGL will sometimes be referred to as "present-day deforestation" or "present-day land cover". The second scenario involved deforestation exclusively on cropland and is referred to as "Deforested Crop Land" (DCL) (Fig. 1b). We use the DCL

scenario to evaluate the potential impact of restoring all grazing land back to its natural state, essentially creating an extreme afforestation scenario, while maintaining the present-day croplands for agricultural food production.

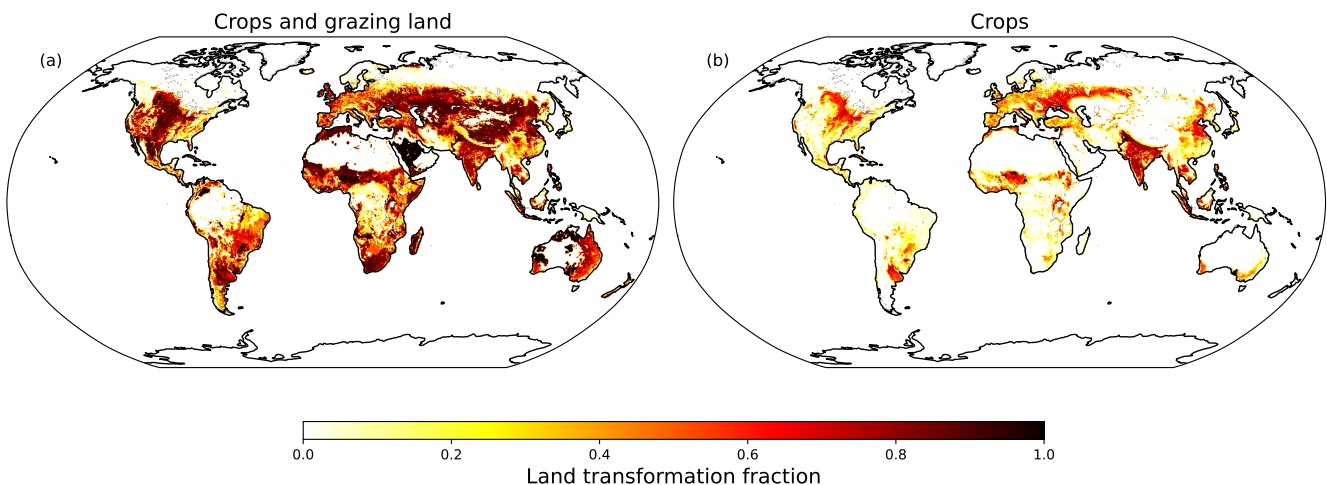

**Figure 1.** Deforestation maps used for the deforesting tree PFT's in LPJ-GUESS. Products derived from HYDE v3.2 based on the year 2015.

All simulations were conducted over 12 years (2000-2011), with the initial 2 years excluded from the analysis to ensure proper spinup and equilibrium state in the analysed data. The vegetation initial states for all simulations were taken from a previous non-chemistry 50-year run under similar atmospheric states. For this study, the simulations were performed in T63L31 resolution, i.e., approximately $1.9° \times 1.9°$ (or approx. $180 \times 180$ km at the Equator) with 31 vertical levels. The meteorological fields were nudged in the troposphere towards ERA-Interim reanalysis data (Dee et al., 2011) for the respective years 2000-2012. Nudging meteorology is important to prevent deviations in our simulations caused by feedbacks related to temperature and dynamics. By implementing nudging, these feedbacks are effectively suppressed, enabling us to evaluate changes in atmospheric states solely due to perturbed BVOC emissions resulting from land cover change.

The corresponding forcing at the sea surface (Sea Surface Temperatures (SSTs) and Sea Ice Coverages (SICs)) is also inferred from the nudging data with continuous variation. Tracers were initialized using climatological data from previous simulations spanning 2000 to 2020, while the $CO_2$ concentrations in the radiation and vegetation schemes were kept fixed at 384 ppmv, representing the year 2015.

Greenhouse gas mixing ratios, including $N_2O$, $CH_4$, $CO_2$, Halon, and $H_2$, were prescribed at the surface level using data from the Chemistry-Climate Model Initiative (CCMI) for the year 2015 (Eyring et al., 2013). Stratospheric $H_2SO_4$ mixing ratios were derived from a time series provided by the CCMI database. Biomass burning emissions were simulated by the BIOBURN submodel, which imports dry matter data from GFEDv4.1s (Randerson et al., 2018) and employs emission factors from Andreae (2019), also based on the year 2015. Anthropogenic emissions of black carbon (BC), carbon monoxide (CO), nitrogen oxides ($NO_x$), organic carbon (OC), sulfur dioxide ($SO_2$), alcohols, and organic gases were based on the Copernicus Atmosphere Monitoring Service (CAMS-GLOB-ANTv4.2 and CAMS-GLOB-AIRv1.1) (Granier et al., 2019). Degassing vol-

canic climatology data were obtained from the AEROCOM project (Dentener et al., 2006). Oceanic emissions and deposition were calculated online using the AIRSEA submodel (Pozzer et al., 2006; Lana et al., 2011; Fischer et al., 2012) for dimethyl

sulfide (DMS), acetone ($CH_3COCH_3$), methanol ($CH_3OH$) (with an under-saturation of 6%), and isoprene ($C_5H_8$). Natural emissions of ammonia ($NH_3$) were based on the GEIA database (Bouwman et al., 1997). Biogenic nitrogen oxide (NO), sea salt (Guelle et al., 2001), and dust emissions (Klingmüller et al., 2018) were calculated online from the ONEMIS submodel (Kerkweg et al., 2006), using corresponding climate states from EMAC. We note that all scenarios (PNV, DCGL, DCL) use the same meteorology and emissions as described above. This means we do not account for past or future atmospheric forcing

on vegetation and BVOC emissions (e.g., temperature changes). Instead, we focus only on the impact of perturbed BVOC emissions in a present climate state.

## 3 Results

### 3.1 Present-day land cover

This section explores changes in the natural land biosphere caused by human deforestation for crops and grazing land, based

on 2015 HYDE land use data. We demonstrate how these alterations in the biosphere propagate to the atmosphere, affecting BVOC surface fluxes and impacting aerosol burden and other atmospheric states.

### 3.1.1 Changes in vegetation states

Fig. 2 shows the PNV scenario's spatial distribution of the vegetation fraction for tree (a) and grass (b) PFTs. The vegetation fraction refers to the area covered by vegetation per unit of ground area (not to be confused with the Leaf Area Index). Panels

c and d show the changes in tree and grass PFTs from the deforestation scenario (DCGL-PNV), respectively. As illustrated in Fig. 2c, the deforestation routine implemented in LPJ-GUESS effectively prevents tree establishment in the transformed regions outlined in Fig. 1a, resulting in the dominance of grass PFTs in these areas (Fig. 2d). Present-day land cover decreases the global tree coverage by 1026 Megahectares (Mha), while it expands the grass coverage by 953 Mha. We find that the global carbon biomass, defined as the amount of functional tissue in land vegetation (including roots), decreases from 567 PgC in the

PNV scenario to 503 PgC in the DCGL scenario. This indicates that the present-day land biosphere has lost 64 PgC compared to natural vegetation (Fig. S1 in the supplementary material).

Here, we acknowledge some limitations in this assessment. LPJ-GUESS simulates only two grass PFTs, C1 and C2, which represent grasslands rather than agricultural crops. While the BVOC emission rates from crops differ from those of natural grasslands (Weber et al., 2023), we contend that this discrepancy is not substantial, as most land use changes are related to

235 grazing rather than crop cultivation. Additionally, in its current configuration, LPJ-GUESS does not simulate shrub PFTs, which means we may also be overlooking some biomass and BVOC emissions from grasslands.

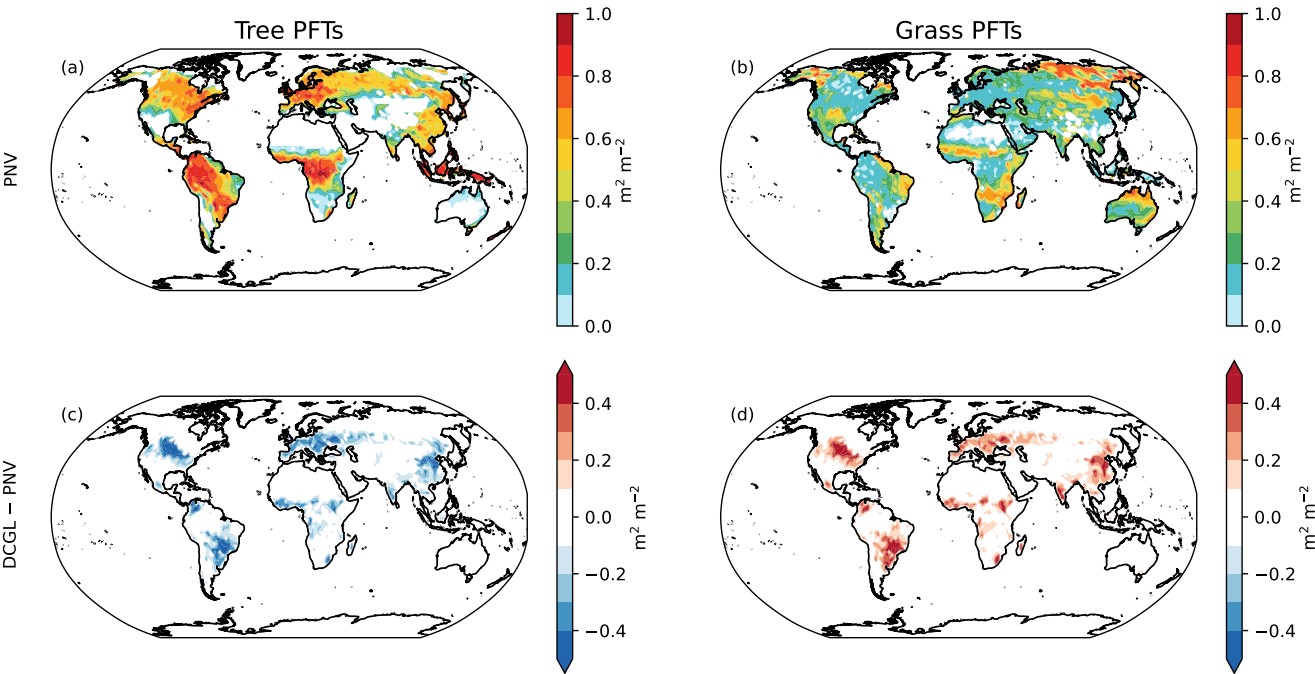

**Figure 2.** Changes in the tree and grass cover fractions (area of covered by vegetation per unit ground area). The right panels show the tree PFTs while the left panels show grass PFTs. Panels a-b show the global distribution of the vegetation fraction in the PNV scenario. Panels c-d show the changes from the deforestation (DCGL−PNV) scenario.

### 3.1.2 BVOC surface emissions

The clearing of biomass from deforestation practices impacts the BVOC global emissions. Fig. 3 a-b shows the surface isoprene and monoterpene emissions in the PNV scenario. The model simulates fluxes of up to 100 mg m$^{-2}$ day$^{-1}$ of isoprene, and 8 mg m$^{-2}$ day$^{-1}$ of monoterpene in the tropics. Fig. c-d shows the spatial changes in the BVOC fluxes resulting from deforestation.

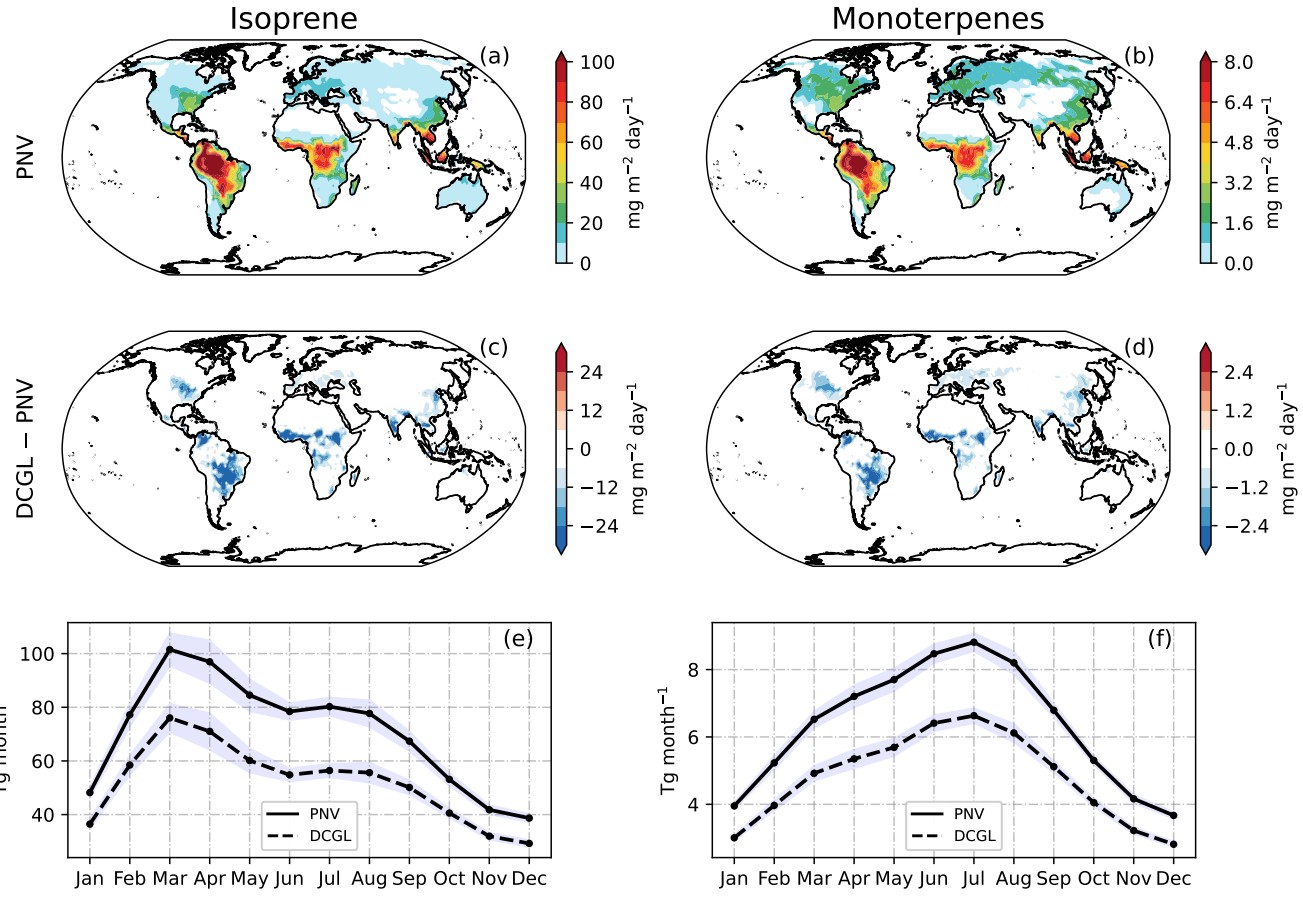

**Figure 3.** Isoprene and monoterpene surface fluxes for the PNV scenario (a-b). Subplots c and d show the spatial difference in isoprene and monoterpene emission fluxes (DCGL − PNV). Monthly emissions based on the 10-year global average for isoprene (e) and monoterpenes (f). The shading represents 1 standard deviation derived from the monthly averages based on 10 simulated years. Fluxes in the Southern Hemisphere were shifted by 6 months to juxtapose the seasonal cycle.

In Fig. 3e-f, the temporal profile of global monthly emission totals is depicted, with shading indicating 1-$\sigma$ variability based on 10 simulated years. To capture the true seasonal cycle, values from the Southern Hemisphere were shifted by 6 months before combining fluxes from both hemispheres. In the PNV scenario, the global annual emission flux for isoprene totalled 845.7 Tg, which decreased to 620.9 Tg in the DCGL scenario, marking a reduction of 224.8 Tg (a 27% decrease relative to PNV). Similarly, for monoterpenes, the annual global emissions in PNV were 76.0 Tg and 57.3 Tg in DCGL, respectively, indicating a decrease of 18.7 Tg (a 25% decrease relative to PNV). Consequently, the global, annual source of BVOCs decreases by 243.5 Tg.

### 3.1.3 Aerosol burden, cloud interactions, and radiative effects

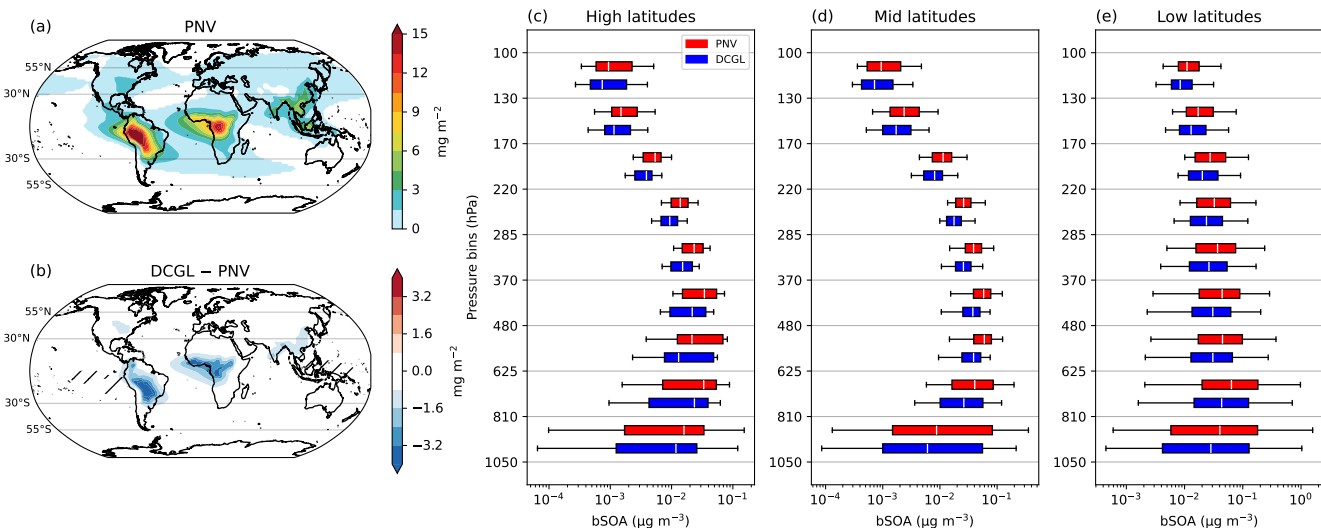

**Figure 4.** Total column bSOA from PNV (a) and changes in bSOA from deforestation (DCGL − PNV) (b). Panels (c), (d), and (e) show the vertical profiles of bSOA, represented by box-whisker plots for different pressure bins. The white line marks the median, the box corresponds to the lower and upper quarterlies, and the whiskers represent the 5th–95th percentile of the spatial mean over the 10 years simulated. The latitude ranges are defined as follows: High latitudes (90–55°S and 55–90°N), mid-latitudes (55–30°S and 30–55°N), and low latitudes (30°S–30°N). A log scale is used for the x-axis of (c), (d), (e). Panel (b) includes diagonal crosshatching to indicate areas that are not statistically significant, based on a two-tailed Student's t-test with a 90% confidence level.

Fig. 4 illustrates the bSOA burden, including the total column burden for the PNV depicted in Fig. 4a, as well as changes in the column burden resulting from deforestation (4b), and, the vertical profiles across three latitude bands (4c-e). Panel (b) includes diagonal crosshatching to indicate areas that are not statistically significant, based on a two-tailed Student's t-test with a 90% confidence level using annual global means. The model shows that most changes in the bSOA column burden are statistically significant, except in certain regions of the tropical Pacific Ocean. The vertical profile is represented using box-whisker plots, showcasing variations at different pressure bins. In the PNV scenario, the bSOA load exhibits its highest concentration over the Amazon forest and the Congo basin, reaching a column mass of up to 15 mg m$^{-2}$. In the PNV scenario, we estimate a total annual OA atmospheric burden of 1.87 Tg, of which 0.56 Tg are bSOA, while in DCGL the total annual OA atmospheric burden reduces to 1.70 Tg, of which 0.40 Tg are bSOA. The bSOA burden decreased by 29% while the total OA decreased by 9%.

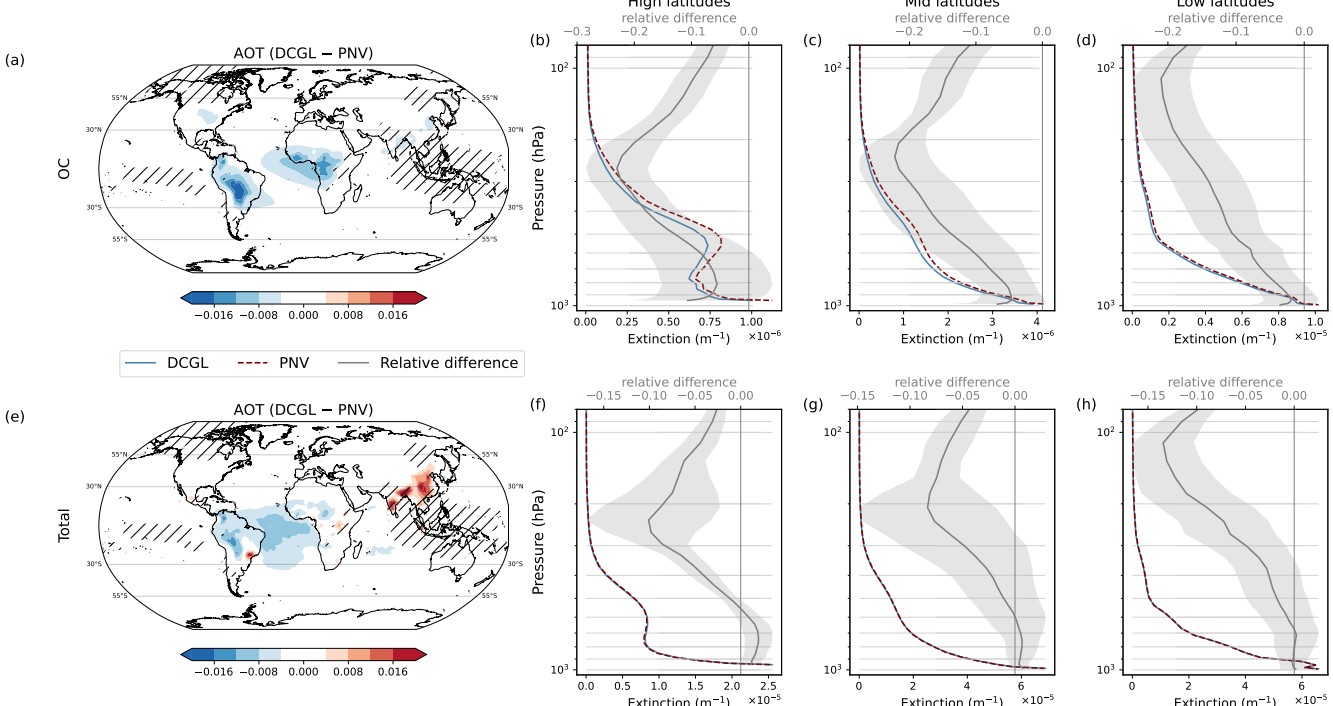

**Figure 5.** Spatial maps of the total column aerosol optical thickness (AOT) at 550 nm for organic carbon (OC) (a), and total aerosol (e). Vertical profiles for the aerosol extinction at 550 nm for DCGL, PNV, along with the relative differences ((DCGL−PNV)/PNV) in different latitude bands. Organic aerosol in (b), (c), (d), and total aerosol in (f), (g), (h). The latitude ranges are defined as follows: High latitudes (90–55°S and 55–90°N), mid-latitudes (55–30°S and 30–55°N), and low latitudes (30°S–30°N). The grey area represents 1 standard deviation of the spatio–temporal mean (grey line). Please note the different scales for the relative differences. Panels (a) and (e) include diagonal crosshatching to indicate areas that are not statistically significant, based on a two-tailed Student's t-test with a 90% confidence level.

Fig. 5 shows changes in the total column aerosol optical thickness (AOT) and extinction at 550 nm for OA (top panels) and for the total aerosols (bottom panels). The aerosol optical thickness (AOT) is determined by integrating the aerosol extinction (m$^{-1}$) over the full atmospheric column, making it a dimensionless quantity. Fig. 5a illustrates the absolute changes in AOT from organic carbon (OC) and suggests a decrease of up to 0.02 over the Amazon forest and Congo basin in Africa. Panels b, c and, d, show a consistent decrease in the OC extinction, with the highest relative difference peaking at around 250 hPa in the high and mid-latitudes, and around 100 hPa in the tropics. Our calculations indicate a global decrease in the OC AOT of −7%, with corresponding decreases of −10%, −9%, and −5% in high, mid, and low latitudes. EMAC simulates the optical properties of six aerosol species, namely organic carbon (OC), black carbon (BC), water-soluble inorganic aerosols (WASO), dust (DU), sea salt (SS), and aerosol-associated water (H2O). The aerosol extinction at 550 nm from the total aerosol suggests a decrease over the tropics (Fig. 5e), however, this trend varies significantly across latitudinal bands. Notably, there is a marginal increase of 0.1% in the high latitudes with declines of −0.5% and −0.6% in mid and low latitudes, respectively. Diagonal

crosshatching in panels (a) and (e) highlights regions where changes in OC and total aerosol extinction are not statistically significant. The model suggests that most changes in AOT are statistically significant, except in regions of the tropical Pacific Ocean and Southeast Asia.

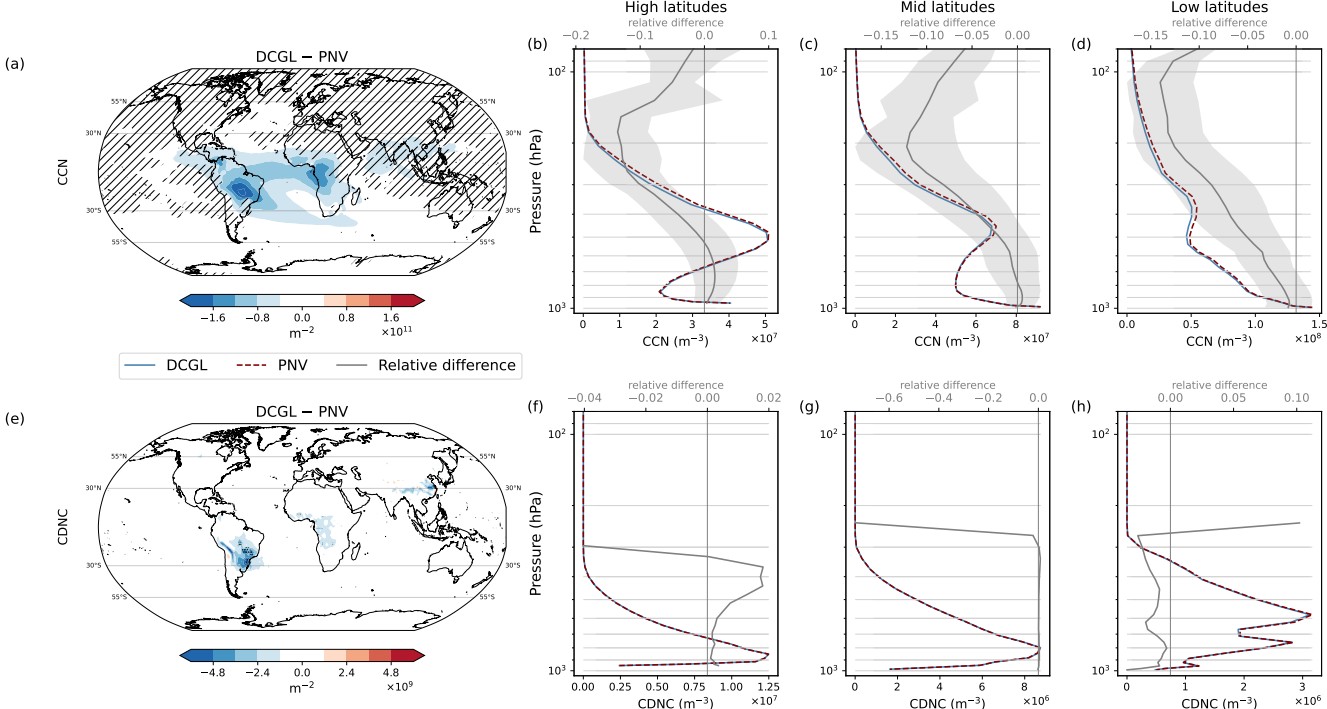

**Figure 6.** Changes in cloud condensation nuclei (CCN) at 0.2% supersaturation (top panels) and cloud droplet number concentration (CDNC) (bottom panels). The panels on the left-hand side, (a) and (e), show the spatial difference in the total column burden (number of particles per square meter), for CCN and CDNC, respectively, emerging from DCGL compared to natural vegetation (DCGL − PNV). The panels on the right-hand side show the total-column vertical profiles from DCGL and PNV simulations and their relative difference are shown. Figure details are the same as in Fig. 5; however, the grey area represents 1 standard deviation of the spatio–temporal mean is only applied in panels (b), (c), (d), and in panel (e), dot hatching is applied to highlight areas with statistically significant differences.

Although this model setup excludes organic NPF, aerosol growth via the condensation of organics can still impact CCN and CDNC by increasing particle size. This growth pushes existing particles closer to the critical radius required for them to act as CCN and activate into cloud droplets, as described by Köhler theory in the activation parameterisation used by the CLOUD submodel in EMAC. Fig. 6 illustrates changes in the CCN at 0.2% supersaturation and CDNC. The reduced aerosol burden in DCGL leads to a CCN decrease of up to $2 \times 10^{11}$ particles per m$^2$ over the tropical regions. Our calculations suggest a global decrease in CCN of 4.8%, with respective declines of 3%, 5%, and 5% in high, mid, and low latitudes, respectively. Diagonal crosshatching in panel (a) indicates that CCN column burden changes are only statistically significant in tropical South America, tropical Africa, and much of the Southern Hemisphere. Panels (b), (c), and (d) show the vertical profile of

CCN for both DCGL and PNV. In high and mid-latitudes, the difference in CCN concentration peaks around 200 hPa, showing a decrease of approximately 12% in the DCGL scenario. In the tropics, the peak difference occurs higher up in the atmosphere (around 950 hPa), also with a decrease of around 12%. Our results indicate that the lower availability of CCN (Fig. 6a) induces a decrease in CDNC in South America, Central Africa and Eastern China. Globally, deforestation leads to a decrease of 0.2% in CDNC. Spatial averages over high and mid-latitudes suggest a general increase of 0.2% and 0.3%, respectively. Conversely, over the tropics, CDNC decreases by 0.6%. Panel (e) uses dot hatching to highlight statistically significant changes in the CDNC column burden. The model indicates that only small regions in South America show statistically significant changes. The smaller signal strength and significance in the CDNC column burden compared to CCN changes arise because most CCN do not coincide with the cloud layer and, therefore, do not interact with clouds. In a separate analysis focusing on the pressure level where the difference in CDNC between the two simulations is maximized, we find statistical significance across the board, occurring on average around 700 hPa, or approximately 3.3 km. This means that at this layer global CCN is influencing cloud droplet numbers. Panels (f), (g), and (h), which display the vertical profiles of CDNC, show that the strongest influence on CDNC, characterised by small but consistent reduction throughout the column, occurs in the tropics. This pattern suggests that shifts in aerosol loading directly affect cloud properties, particularly in regions influenced by perturbed organic aerosols from BVOC precursors. However, the magnitude of the effect appears small. Additionally, the standard deviation in relative changes exceeds 1000%, and thus, it was excluded from Fig. 6f-h to prevent distorting the scale. This high standard deviation highlights the spatial variability across latitudinal bands and suggests that the observed changes are spatially heterogeneous.

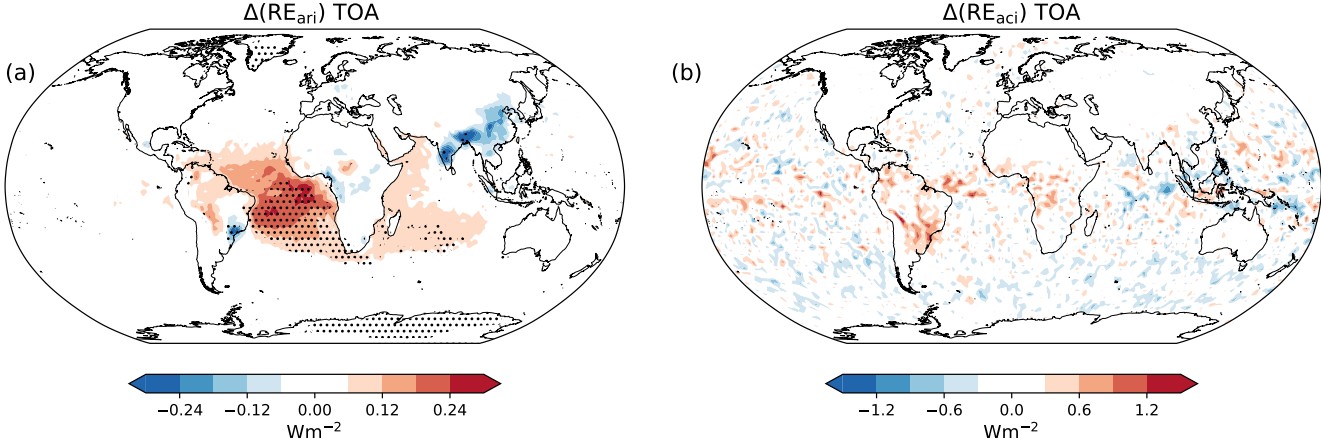

**Figure 7.** Aerosol and cloud radiative effect mediated by aerosol changes from deforestation (DCGL−PNV). Panels a and b show the top-of-the-atmosphere (TOA) direct (ARI) and indirect (ACI) radiative effect, respectively. Dot hatching is applied to highlight areas with statistically significant differences, based on a two-tailed Student's t-test with a 90% confidence level.

Fig. 7 shows the changes on $RE_{ari}$ and $RE_{aci}$ resulting from the conversion of natural vegetation to crop and grazing land. We estimate a global increase in aerosol radiative effect ($RE_{ari}$) of 60.4 mW m$^{-2}$, with the tropics showing a notably higher increase, averaging 91.2 mW m$^{-2}$. Most of the $RE_{aci}$ is positive in tropical regions due to the reduced aerosols. In contrast, the

increased aerosol extinction observed over Southeast Asia (Fig. 5e) corresponds to a negative, or cooling, effect in that area. The model indicates a less prominent signal in cloud radiative effect ($RE_{aci}$), with a noticeable uptick of $+27.5$ mW m$^{-2}$ over the tropics, but a minor global cooling effect of $-8.7$ mW m$^{-2}$. The simulations indicate that the signal in REari is significant over the tropical Atlantic Ocean, parts of South America, and India. However, the model suggests that REaci changes are not

statistically significant. Fig. S6 in the supplementary material include maps with relative changes in bSOA, AOT, CCN, CDNC, and RE resulting from present-day land use cover compared to natural vegetation cover.

## 3.2 Grazing land restoration

In this section, we evaluate the changes resulting from the restoration of all grazing land to natural vegetation. In Section 3.1, PNV was used as the baseline scenario to assess changes from present-day land use cover (DCGL). In this following section,

DCGL serves as the baseline scenario, with the sensitivity run being the DCL with only deforested crop cultivation, as shown in Fig. 1b. These analyses therefore represent an extreme reforestation scenario.

### 3.2.1 Changes in vegetation and BVOC emissions

In this scenario, where it is assumed that all grazing land is restored to natural vegetation (DCGL$-$DCL), the global tree cover increases by 600 Mha compared to present-day land use cover (Fig. S2 in the supplementary material). We also find that in this

scenario, the global biomass increases to 546 PgC, which is an increase of 43 PgC compared to present-day land cover. Fig. S1 in the supplementary material shows the spatial distribution of biomass per unit land area and the shifts resulting from the land use scenarios considered here.

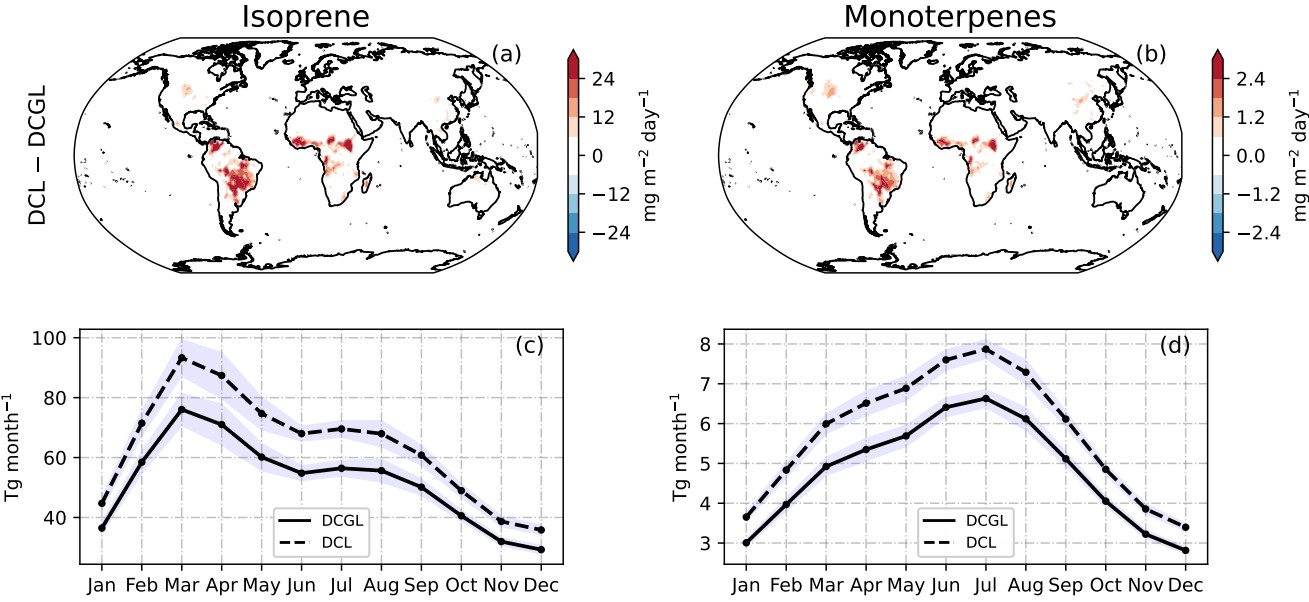

**Figure 8.** Spatial difference in isoprene and monoterpene emission fluxes (DCL−DCGL) (a-b). Monthly emissions based on the 10-year average for isoprene (c) and monoterpenes (d). The shading represents 1 standard deviation derived from the monthly averages based on 10 simulated years. Fluxes in the Southern Hemisphere were shifted by 6 months to conserve the seasonal cycle.

Fig. 8 depicts the changes in surface isoprene and monoterpene emissions from the grazing land restoration run compared to present-day land cover. These emission changes align with the alterations in vegetation cover (Fig. S2). Globally, annual isoprene emissions in the DCL scenario increase by 140.4 Tg (+23%) relative to the DCGL scenario, while monoterpene emissions increase by 11.6 Tg (+20%).

### 3.2.2 Changes in atmospheric states

Fig. 9 summarises the changes in atmospheric states following the conversion of grazing land to reforestation (DCL − DCGL) including radiative effects from aerosols and cloud properties (Fig. 9e-f). The column mass of bSOA (Fig. 9a) experiences a notable rise of over 3 mg m$^{-2}$ across tropical South America and Central Africa (Fig. S3b). This rise increases the bSOA burden from 0.40 Tg in the baseline scenario to 0.50 Tg (+0.10 Tg, +25.7%), consequently perturbing the total OA burden, which increases from 1.70 Tg to 1.80 Tg (+0.11 Tg, +6.3%). Fig. S3 in the supplementary material includes the vertical bSOA profiles at the three latitude bands. The observed changes are statistically significant, except in regions over the tropical Pacific Ocean and Southeast Asia. The global mean OC AOT at 550 nm increases by 3.8%, with corresponding increases of 5.5%, 5%, and 2.4% in high, mid, and low latitudes, respectively. The global mean AOT at 550 nm (Fig. 9b) from the total aerosols increases by 0.3%, with a corresponding decrease of 0.1% in the high latitudes and an increase of 0.4% in both mid and low latitudes. Statistically significant changes in AOT at 550 nm occur only in certain regions of the tropical Atlantic Ocean and

parts of South America (see Fig. 9b). Spatial maps of the AOT and extinction vertical profiles for the different latitude bands are shown in Fig. S4.

In this scenario, we estimate a global increase in CCN of 3.5%, with increases of 2.5%, 3.7%, and 3.7% in high, mid, and low latitudes, respectively. Statistically significant changes occur in tropical South America, the tropical Atlantic Ocean, and most of the Southern Hemisphere (Fig. 9c). CDNC increases globally by 0.2%, with decreases of 0.1% in high and mid-latitudes, and an increase of 0.5% in the tropics. Statistically significant changes only occur in a small part of South America (see Fig. 9d). Spatial maps and vertical profiles for the different latitude bands for CCN and CDNC can be found in Fig. S5 of the

supplementary material. The global ARI effect ($RE_{ari}$) is $-38.2$ mW m$^{-2}$, while the radiative effect from changes in cloud properties ($RE_{aci}$) is found to be $+1.6$ mW m$^{-2}$. While REari is statistically significant over some parts of the tropical Atlantic Ocean, REaci changes are not statistically significant.

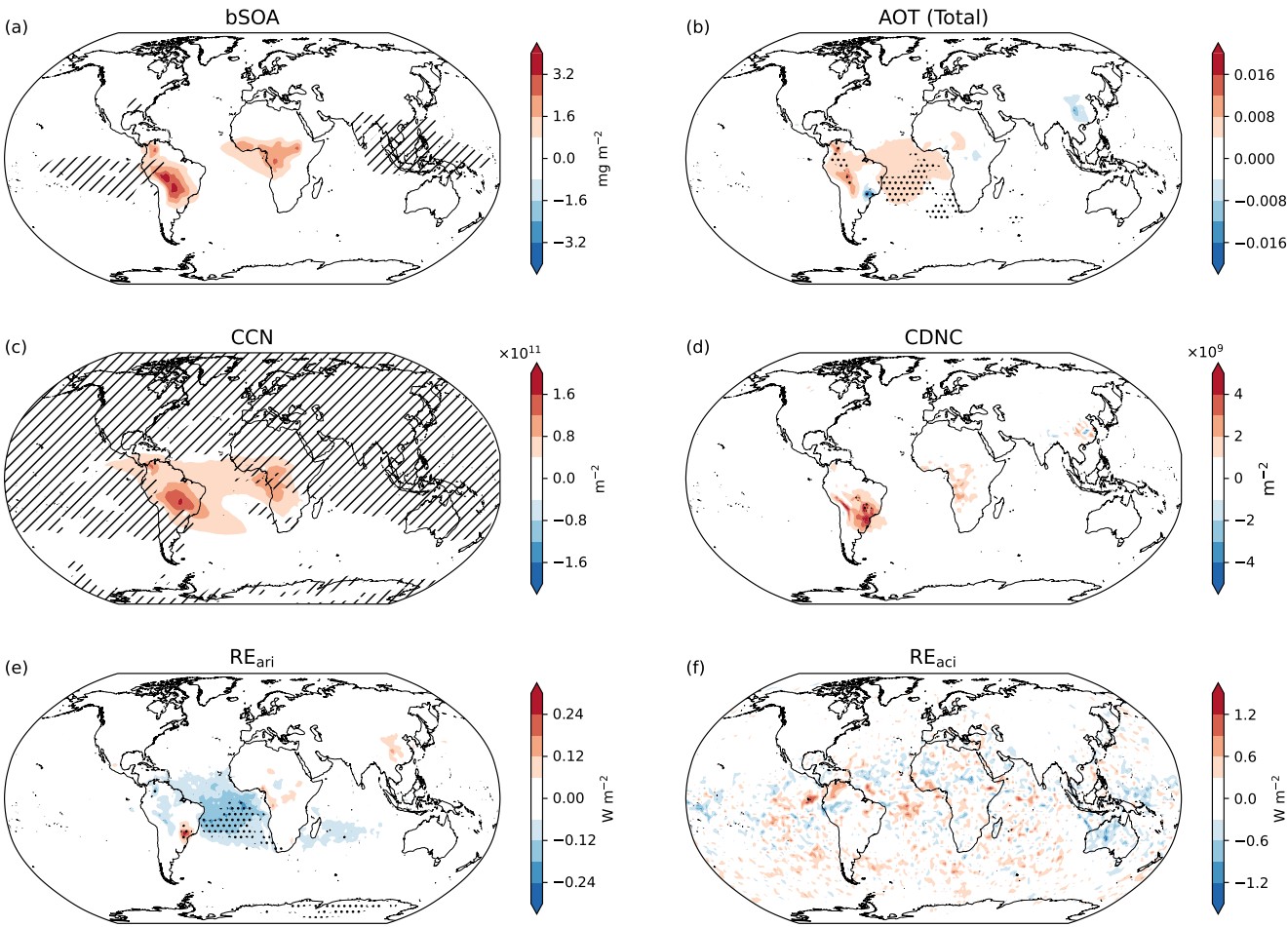

**Figure 9.** Absolute changes in atmospheric states resulting from restoring present-day grazing land. Maps show variations in; (a) bSOA column mass, (b) total aerosol optical thickness (AOT), (c) cloud condensation nuclei (CCN), (d) cloud droplet number concentration (CDNC), (d) aerosol radiative effect, and (e) cloud radiative effect. Diagonal crosshatching is applied in panels (a) and (c) to indicate areas that are not statistically significant, while dot hatching is used in panels (b), (d), (e), and (f) to highlight areas with statistically significant differences, based on a two-tailed Student's t-test with a 90% confidence level.

Table 1 outlines the changes in vegetation and atmospheric variables in the two land cover scenarios considered in this study: present-day land cover versus natural vegetation (DCGL-PNV), representing deforestation, and restoration of grazing land versus present-day land cover (DCL-DCGL), representing reforestation. The results indicate that both deforestation of natural vegetation to the present-day land cover, and reforestation of present-day grazing lands significantly impact vegetation and atmospheric variables.

|  | DCGL−PNV | | DCL−DCGL | |
| --- | --- | --- | --- | --- |
|  | Abs. | Rel. | Abs. | Rel. |
| Tree cover | −1026 Mha | −18% | +600 Mha | +13% |
| Veg. Biomass | −64 PgC | −11% | +43 PgC | +9% |
| Isoprene | −224.8 Tg | −27% | +140.4 Tg | +23% |
| Monoterpenes | −18.7 Tg | −25% | +11.6 Tg | +20% |
| bSOA | −0.16 Tg | −29.0% | +0.11 Tg | +25.7% |
| OA | −0.17 Tg | −9.3% | +0.12 Tg | +6.3% |
| AOT(OC) | $-8.0 \times 10^{-4}$ | −7% | $+4.4 \times 10^{-4}$ | +3.8% |
| AOT(Total) | $-3.0 \times 10^{-4}$ | −0.5% | $+2.5 \times 10^{-4}$ | +0.3% |
| CCN | $-2.6 \times 10^{10}$ m$^{-3}$ | −4.8% | $+1.7 \times 10^{10}$ m$^{-3}$ | +3.5% |
| CDNC | $-4.7 \times 10^{7}$ m$^{-3}$ | −0.2% | $+3.3 \times 10^{7}$ m$^{-3}$ | +0.2% |
| RE$_{\text{ari}}$ | +60.4 mW m$^{-2}$ | +2.2% | −38.2 mW m$^{-2}$ | −1.4% |
| RE$_{\text{aci}}$ | −8.7 mW m$^{-2}$ | −0.002% | +1.6 mW m$^{-2}$ | +0.002% |

**Table 1.** Changes in vegetation and atmospheric variables for the two land cover scenarios: present-day land cover vs. natural vegetation (DCGL−PNV) and restoration of grazing land vs. present-day land cover (DCL−DCGL). Tree cover, vegetation biomass, isoprene, monoterpenes, bSOA, and OA are global yearly sums, while AOT, CCN, CDNC, and RE, are global yearly means.

## 4 Discussion

The present study offers insights into the ongoing impact of human-induced deforestation and potential reforestation on atmospheric aerosols, building upon existing literature. In this study, a decline or increase in vegetation was assumed to drive changes in BVOC emissions and their impacts on atmospheric aerosols. This assumption can be validated by comparing the calculated decline in tree cover and vegetation biomass over the years from the vegetation model with values from the literature. For instance, Hu et al. (2021) estimates that between 1992 and 2018, 722 Mha of forests were converted into agricultural land, while Bhan et al. (2022) estimates the terrestrial biosphere's global carbon stock for 1950 to be 450 PgC. These estimates align with the values simulated by LPJ-GUESS, suggesting that the changes in the biosphere due to land use changes incorporated in this study are consistent with earlier findings.

### 4.1 BVOC emissions changes

Studies agree that land use changes predominantly affect BVOC emissions through the expansion of croplands in tropical regions such as the Amazon, central Africa, and Southeast Asia (Ganzeveld et al., 2010; Lathière et al., 2010; Hantson et al., 2017; Szogs et al., 2017). For example, Lathière et al. (2006) show a reduction of approximately 24% in isoprene emissions between 1901 and 2002. Additionally, Unger (2014) reports that land cover changes from the 1850s to the 2000s resulted in a global decrease of approximately 35% in BVOC emissions. In a comprehensive literature review exploring the influence

of Land Use and Land Cover Change (LULCC) on atmospheric composition and climate, it was estimated that from the preindustrial era to the present, LULCC has led to a decrease of 15-36% in global isoprene emissions (Heald and Spracklen, 2015). Scott et al. (2018) reports a global decrease in isoprene and monoterpenes emissions of 87% and 94%, respectively, resulting from complete global deforestation. The 26% decrease in BVOC emissions reported in this study is in line with the findings reported in the literature, particularly because many studies analyse changes from preindustrial times, while our work focuses on changes from human land use compared to natural vegetation. Although cropland and grazing land expansion was not as significant as during or after the industrial era, it did exist on a smaller scale. Attention must also be given when comparing studies with and without temperature feedback, given the strong dependence of BVOC emissions on temperature. In this work, similar to the calculations in Scott et al. (2018), temperature feedbacks were suppressed.

The global yearly isoprene and monoterpene emission budgets from this study are relatively high compared to other global estimates in the literature, as shown in Vella et al. (2023a). Some atmospheric chemistry studies using MEGAN in EMAC, e.g., Pozzer et al. (2022a), employ a global scaling factor to dampen the global emissions to desired values. However, in this study, no scaling factors were applied, which means that the values reported here may be slightly overestimated.

## 4.2 Aerosol burden

The rapid oxidation of BVOC, yields oxygenated intermediate species near the surface, which act as precursors for the formation of bSOA. This explains the high abundance of bSOA in the lower atmosphere (see vertical profiles in Fig. 4). As bSOA is dispersed in the atmosphere, concentrations typically decrease due to mixing and dilution. However, the vertical profile of bSOA is also governed by the vapour pressure of the oxygenated intermediates, which decreases with lower temperatures as they ascend in the troposphere. When they are transported upward, the reduced volatility at lower temperatures prompts the transition of oxygenated intermediates into the aerosol phase, contributing to the formation of bSOA. Consequently, this interplay between oxidation, transport, and vapour pressure-driven processes imparts a distinctive D-shaped vertical profile of biogenic SOA concentrations, with elevated concentrations near the surface, followed by an increase in bSOA towards higher altitudes, and ultimately a decline in the upper atmosphere. In the low latitudes (tropics), the D-shape in bSOA concentration is only faint (Fig. 4e). The prevalence of warm air, relatively high boundary layers, and the occurrence of deep convection transporting both aerosol particles as well as precursors into the upper troposphere lead to the maximum gas-to-particle partitioning occurring at higher altitudes, typically around 100-200 hPa, resulting in a secondary local upper tropospheric enhancement.

Several studies estimate the global mean burden bSOA at ∼0.5-0.77 Tg (Henze et al., 2008; Pye et al., 2010; Hoyle et al., 2007; Tsigaridis and Kanakidou, 2007; Tilmes et al., 2019). This indicates that in the present-day land cover scenario, ORACLE provides a lower estimate of bSOA burden (0.40 Tg) compared to literature values. Pozzer et al. (2022b) evaluated OA from ORACLE in EMAC and found that surface concentrations are well represented, however, OA is strongly underestimated in the free troposphere. The work from Pozzer et al. (2022b) employed the Mainz Organic Mechanism (MOM), which is a more complex chemical mechanism compared to the one used in this study (i.e., the Mainz Isoprene Mechanism, MIM), nevertheless, this indicates that even with a more complex mechanism EMAC calculates a lower OA globally. Heald and Geddes (2016) reports a global annual mean tropospheric burden of bSOA to decrease by 13% due to land use change between 1850 and

2000, while Scott et al. (2018) estimates a 91% decrease in SOA from simulated global deforestation. Our model calculations suggest a reduction of approximately 30% in bSOA from crop and grazing land deforestation compared to natural vegetation.

Regionally, the derived total AOT exhibits opposing effects, particularly, pockets of increased total AOT are evident in confined regions and notably in Southeast (SE) Asia (Fig. 5e). Here, the extinction from H2O and WASO is found to be increasing, effectively masking the expected reduction in OA extinction from lower BVOC precursors in DCGL. This phenomenon is linked to the growth of aerosol particles into what is commonly referred to as the "Greenfield gap" - a range characterised by lower deposition velocities and scavenging values (Greenfield, 1957). The extended atmospheric lifetime of particles in this region adds to the burden of WASO compounds and their associated water uptake, thereby amplifying aerosol extinction. Moreover, the absence of organics exacerbates this effect, leading to a disproportionate condensation on accumulation mode particles at the expense of Aitken mode particles. Consequently, the increased presence of water-soluble compounds, coupled with increased aerosol water content and a shift in size distribution towards larger particles, collectively contribute to the observed increase in extinction in these specific areas. While both factors contribute, enhanced scattering is the dominant influence on aerosol optical thickness. Nevertheless, plotting the relative difference of total AOT (Fig. S6b) shows that this effect is not very prominent, with an increase in AOT in this region of less than 4%. This phenomenon occurs only in DCGL−PNV. In DCL−DCGL, the increase in organic aerosols allows condensation on Aitken mode particles, limiting growth in the accumulation mode, which aligns with our explanation.

### 4.3 Cloud properties and radiative effects

Studies focusing on changes in cloud properties due to perturbed BVOC precursors and SOA loading resulting from land use change are somewhat limited. Scott et al. (2014) investigated the impact of bSOA on surface CCN and CDNC at cloud height (approximately 900 hPa) in the present-day atmosphere, and found that bSOA increases the mean annual concentration of CCN by 3.6–21.1%, and the global annual mean concentration of CDNC by 1.9–5.2%. In this work, we show that present-day deforestation, when compared to the potential vegetation scenario, yields a decrease of 4.8% in the total column CCN, but only a small decrease of 0.2% in column CDNC burden, with a slightly more prominent reduction of 0.6% over tropical regions. The most significant change in cloud droplet count occurs over the Amazon, with an approximate 8% decrease in this region (Fig. S7d). The changes in CDNC simulated by EMAC, however, seem to be very localised, in particular over the Amazon rain forest, but the global impact on CDNC is small. Deforestation-induced changes in CDNC are less than 10% over the Amazon and less than 1% globally (Fig. S6 and S7).

It is worth pointing out that the effect on cloud droplet numbers reported here only stems from aerosol-cloud interactions. In reality, reduced tree cover may lead to less evapotranspiration, which modifies the Bowen ratio (sensible to latent heat ratio) and potentially influences the cloud effect, however, this feedback is suppressed in our simulations. This is in addition to roughness changes and albedo changes that can also influence cloud cover (Cerasoli et al., 2021). Moreover, in this model setup, only aerosol-cloud interactions with large-scale clouds are considered. Therefore, the impact of changes in the aerosol burden on convective clouds and their potential radiative effects are not captured. The lack of organic NPF descriptions in the model may also underestimate the real influence of bSOA on CDNC and REaci reported in this work.

Biogenic SOA is known to have net cooling effects on the Earth's climate by scattering a portion of incoming solar radiation back into space (Zhu et al., 2019; Sporre et al., 2020; Tilmes et al., 2019). Therefore, changes in aerosol numbers and composition from deforestation result in net warming from the lack of aerosol scattering, while reforestation leads to the opposite effect with a net cooling effect (Fig. 7a and Fig. 9e). Scott et al. (2018) showed that full deforestation results in a radiative effect of 117 mW m$^{-2}$ due to aerosols and 200 mW m$^{-2}$ due to clouds. In contrast, projections for deforestation under the RCP8.5 scenario for 2100 indicate that aerosols contribute only 6 mW m$^{-2}$ of radiative effects, with negligible aerosol-radiative interaction (ARI) Scott et al. (2018). Furthermore, O'Donnell et al. (2011) estimated the total secondary organic aerosol (SOA) aerosol-cloud interaction (ACI) effect to be 310 mW m$^{-2}$.

The comparatively small ARI effect reported here from deforestation relative to the PNV scenario (60.4 mW m$^{-2}$) may stem from differences in methodology compared to other studies, which focus on the effects of full deforestation or total SOA (not bSOA). However, our RE$_{ari}$ estimates might be underestimating the actual impact due to comparatively lower bSOA yields in our model runs. The radiative forcing from BVOCs is strongly influenced by SOA yields, which can vary considerably with environmental conditions. SOA yields are sensitive to factors such as temperature and oxidant concentrations, which can alter the overall aerosol burden and their radiative properties (Sporre et al., 2020). Furthermore, the role of oxidant chemistry is crucial in determining the oxidation pathways and the formation of SOA. Variations in oxidant levels, particularly ozone and hydroxyl radicals, can substantially affect SOA formation rates and chemical composition (Weber et al., 2022). A more comprehensive analysis of the chemical processes influencing SOA yields and their associated radiative forcing should be further explored using the EMAC model.

For the afforestation scenario, we acknowledge that this is an idealistic sensitivity analysis, perturbing only the land use cover while maintaining present-day values for greenhouse gas concentrations and anthropogenic emissions. In different future scenarios, these key variables would play a crucial role in determining aerosol radiative effects. This is also highlighted in this study, where we showed that over Southeast Asia, where anthropogenic aerosols are present in relatively high numbers, the reduction in global OA from deforestation resulted in changes in the aerosol-size distribution over this area leading to an opposing effect in aerosol extinction.

This work includes an analysis of the statistical significance of changes in atmospheric states resulting from perturbed surface BVOC emissions due to land use change. In this analysis, we applied a two-tailed Student's t-test with 90% confidence ($p < 0.1$). The t-test was performed on annual means to minimise noise from annual internal variability. We show that changes in surface BVOC emissions lead to a statistically significant bSOA column mass burden on a global scale, except for a portion of the tropical Pacific Ocean. Changes in AOT attributed to OA and total aerosol exhibit statistically significant differences overall, except in Southeast Asia. CCN changes predominantly occur in the tropics (excluding Southeast Asia) and the Southern Ocean. Changes in the CDNC column burden are largely not statistically significant, but CDNC changes at approximately 700 hPa do exhibit statistically significant differences. While REari exhibits a statistically significant signal over the tropical Atlantic Ocean, where most of the changes occur, REaci did not demonstrate statistical significance related to deforestation.

While some studies, such as those employing the UK Earth System Model (UKESM) and the Community Earth System Model (CESM2), indicate statistically significant ARI effects from deforestation at a 95% confidence level (Weber et al.,

2024), other studies, such as the one from Unger (2014), employing the Yale-E2 global carbon–chemistry-climate model, suggest that the global-scale impacts of cropland expansion on bSOA are not statistically significant relative to interannual variability in climate. We argue that the land cover changes assessed in EMAC significantly impact global aerosol budgets. However, it appears that the influence of total cloud droplet numbers and RE$_{aci}$ on a global scale are likely to be minor. Here we emphasise that the statistical significance of a result is not a property of the effect (whether it is large or small) but rather a measure of whether it can be detected (i.e., how likely it is that the result occurred by chance). Our results suggest that the impact on clouds and the corresponding radiative forcing is not significant; however, this is highly dependent on the model setup. We are limited by a 10-year simulation output, which may not be sufficient for a signal in clouds to emerge, given the natural noise and internal variability of the climate system.

## 5 Conclusions

Understanding how land use changes alter atmospheric composition is crucial for grasping the impact of human activities on the Earth system, both past and future. This study presents a comprehensive evaluation of two land-use scenarios, focusing on how changes in BVOC emissions influence the atmosphere and climate. We find that present-day deforestation results in a 73% decrease in isoprene emissions and a 75% decrease in monoterpene emissions. Consequently, the bSOA burden decreases from 0.56 Tg to 0.40 Tg, and the total OA burden decreases from 1.80 Tg to 1.70 Tg. As expected, this leads to a reduction in OC extinction, especially in the tropics near the source of bSOA and its precursors. Upon examining the impact on total aerosol extinction, the model indicates declines in AOT over South America and the tropical Atlantic Ocean. However, aerosol extinction increases over SE Asia in the present-day deforestation scenario compared to PNV conditions. Particularly in SE Asia, the reduction in organic material results in the growth of water-soluble inorganic compounds (WASO). The absence of organics leads to more condensation on accumulation mode particles and less condensation on Aitken mode particles, resulting in increased aerosol water uptake and a shift in the aerosol size distribution towards larger particles. This effectively increases aerosol extinction.

The perturbations in aerosols resulting from deforestation lead to a positive radiative effect in the tropics (warming), stemming from the reduced aerosol burden. The increased presence of WASO compounds over SE Asia contributes to a negative radiative effect (cooling) in this region. However, our findings indicate a global positive radiative effect of 60.4 mW m$^{-2}$ (aerosol + cloud). In the reforestation scenario, isoprene and monoterpenes fluxes increase by 23% and 20% respectively, leading to an increase in the bSOA burden by 0.23 Tg and a negative radiative effect of 38.2 mW m$^{-2}$.

*Code and data availability.* The Modular Earth Submodel System (MESSy, doi:10.5281/zenodo.8360186) is continuously further developed and applied by a consortium of institutions. The usage of MESSy and access to the source code is licenced to all affiliates of institutions which are members of the MESSy Consortium. Institutions can become a member of the MESSy Consortium by signing the MESSy Memorandum of Understanding. More information can be found on the MESSy Consortium Website (http://www.messy-interface.org). The model outputs

relevant to this study are permanently stored in the Zenodo repository and are accessible via https://doi.org/10.5281/zenodo.13906983. Analysis scripts are available upon request from the corresponding author.

*Author contributions.* RV, HT, and MF implemented and tested the deforestation routine. RV prepared the model setup and performed the simulations with inputs from HT, AT, and AP. AT provided technical assistance for ORACLE and provided scripts to handle its output. RV evaluated the simulations, analysed the model results and wrote the article. The results were interpreted by all co-authors, with a special focus on vegetation analysis provided by MF and TH. JL and HT supervised the project. All authors discussed the results and contributed to the review and editing of the paper.

*Competing interests.* At least one of the (co-)authors is a member of the editorial board of *Atmospheric Chemistry and Physics*. The peer-review process was guided by an independent editor, and the authors also have no other competing interests to declare.

*Acknowledgements.* The model simulations have been performed at the German Climate Computing Centre (DKRZ) through support from the Max Planck Society and the JGU Mainz. This work was supported by the Max Planck Graduate Center with the Johannes Gutenberg-Universität Mainz (MPGC). AP acknowledges the European Commission Horizon Europe project FOCI (Grant Agreement No 101056783). HT achknowleges funding from the Deutsche Forschungsgemeinschaft (DFG, German Research Foundation) – TRR 301 – Project-ID 428312742.

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
