# Peer review of "Land cover change influence on atmospheric organic gases, aerosols, and radiative effects"

_EGUsphere, 2024_

## Author Comment (AC1)

Author's response to comments from Anonymous Referee #1:

**"Land use change influence on atmospheric organic gases, aerosols, and radiative effects"**

by Ryan Vella et al.

5   We thank editor and referees for taking the time to review our manuscript and for the valuable feedback. Here, the comments from Anonymous Referee #1 (from August 06, 2024) are reproduced in black, while our comments are presented in blue.

**From Anonymous Referee #1's response:**

This study investigates the impact of anthropogenic land use change in the present day (2015) through
10   perturbed emission of biogenic volatile organic compounds (BVOCs). They compare three simulations: with only natural vegetation, with present day deforestation and present-day deforestation only on crop land, while grass land is reforested. They find that present day deforestation leads to a decrease of 29% in biogenic secondary organic aerosol (SOA) compared to natural vegetation, and that their reforestation scenario (only crop land deforested) lead to an increase 26% compared to present day. In terms of the
15   resulting radiative effect, they find a fairly small warming (0.06W/m2) as a result of deforestation and a similarly small cooling for potential reforestation (-0.038 W/m2).

The study is interesting, and mostly well written and it fits well within the scope of ACP. I do have a fair amount of comments, but most of them are about clarity in the text and the analysis/methods.

20   Thank you for your comments and thorough feedback. We acknowledge that the concerns raised here are valid and the recommended amendments greatly improved our manuscript. Please find our detailed responses below.

My main comments are:

1.There is no estimation of the significance of the results in the results section. This should be fairly easy
25   to fix. I suggest adding shadings to all figures where results are not significant, but it will be especially helpful for interpreting the cloud results I believe. There is a discussion of the significance of the results in the discussion section, but it is not clear what has been done and results are presented only for some properties. See more comments on this below.

Figures now include hatching indicating statistically significant changes. We used a 2-tailed student test with a 90% confidence interval on the annual means.

2. The study does not seem to take into account any changes in surface albedo from land use change and it is not entirely clear to me if BVOC emissions are allowed to affect oxidant chemistry and thus ozone and methane concentrations. I would assume not, since this is not discussed in the paper. The study would benefit, in my opinion, from highlighting early on and in the discussion and conclusion, which effects are and which are not included.

In this work we focus only on changes in the aerosol burden and the associated ARI and ACI. We do not consider changes in surface albedo, roughness length, or feedbacks within the hydrological cycle.

Isoprene and monoterpene emissions are treated as active tracers, where their oxidation can lead to condensation and the formation of SOA. These calculations are conducted within the ORACLE submodel using a volatility basis set (VBS) framework. This approach allows the oxidation of BVOCs to influence oxidant chemistry, and thus ozone and methane concentrations. This study focuses on the aerosol burden from perturbed biogenic SOA, with a detailed analysis of changes in oxidant chemistry to be addressed in a follow-up study. Clarifications regarding the model runs were added earlier in the text.

3. The manuscript lacks information on SOA mass yields, new particle formation mechanisms and early particle growth. Both of these are important for interpreting the results in my view.

NPF is treated only in GMXe through $H_2SO_4$ nucleation (). Organics do not contribute to NPF but condense. This process is simulated with a VBS framework (ORACLE), where volatilities are governed by the oxidation of organic precursors.

ORACLE treats SOA mass yields for different lumped VOCs at varying saturation concentrations ($C^*$) in µg m$^{-3}$ at 298 K based on an assumed particle density of 1.5 g cm$^{-3}$. Isoprene exhibits mass yields across the range of $C^*$, with a peak of 0.03 at 10 µg m$^{-3}$ before declining sharply to 0.015 at 100 µg m$^{-3}$ and reaching zero at 1000 µg m$^{-3}$. Monoterpenes show significantly higher yields, starting at 0.107 for low $C^*$ (1 µg m$^{-3}$) and peaking at 0.600 for higher $C^*$ (1000 µg m$^{-3}$), highlighting the much greater contribution of monoterpenes to SOA formation, especially under conditions of higher saturation concentrations. SOA mass yields in ORACLE are detailed in Tsimpidi et al. (2014).

The gas-phase mechanism (MIM1) includes the following BVOC oxidation pathways: isoprene + OH, isoprene + $O_3$, and monoterpene oxidation (lumped species) with OH, $O_3$, $NO_3$, and $O(^1D)$. This information has been incorporated into the manuscript.

I have the following minor comments:

Abstract: The abstract is on the long side and in my view contains too much background. I would reduce the amount of detail and focus more on results.

Some background information was removed making the abstract more concise.

L8: Is it correct to call it interactive vegetation if it doesn't affect climate variables? Ref what is written in L136.

We thank the reviewer for pointing this out. The only feedback to the atmosphere is via BVOC emissions. For clarity, the text in the abstract was updated as follows:

"In this work, a global atmospheric chemistry-climate **coupled with a dynamic global vegetation model** was employed to study the impacts of perturbing the biosphere..."

L9: "Given that ..." seems a bit like the reader should know this already. I suggest to simplify this part of the abstract and only present what the simulations are meant to represent.

Sentence rephrased as suggested.

L29: There seems to be two versions of the same reference (L437-444) in the manuscript: Forester 2007a and b are the same?

Yes, thanks for seeing this. Manuscript amended.

L44: "... new particles, thus influencing SOA formation significantly". I find this sentence confusing because SOA formation usually would indicate mass which is usually dominated by condensation. I would maybe rather say that the climate impact can be very different due to the impact on aerosol number and thus CCN.

This bit of text "thus influencing SOA formation significantly" was replace by a new sentence: "As a result, monoterpene precursors may have distinct climate impacts given their influence on the aerosol numbers."

L45: A 50 nm limit is too simplified both because it depends on a range of factors when a particle can act as a CCN and because the size at which it interacts efficiently with radiation directly is much higher. Please rephrase.

Agreed. Text is now rephrased.

L47: Forester et al (2007) is the 4th assessment report and the terminology has changed in the two last reports (5th and 6th). I suggest updating to aerosol radiation interactions (ARI) and aerosol cloud interactions (ACI) – especially because these terms are used later in the manuscript.

Text amended as suggested.

L51: "potential natural vegetation" should be explained here.

PNV definition now included.

L52: Should it not say that this is a one way coupling in terms of everything except BVOCs? Or have I misunderstood the set up?

Correct. A sentence was added for clarification.

L59: What does it mean that the model represents atmospheric processes and their interactions with anthropogenic activities? Is there a two-way coupling there?

"anthropogenic activities" was replaced by "human activities". This means that EMAC can be used to study the atmospheric impacts of anthropogenic activities, such as direct emissions and land use change. EMAC has also been used to assess health impacts from atmospheric pollution, e.g., https://www.nature.com/articles/nature15371

L64: I am not sure what details are in Jöckel et al. (2016) and which are also detailed below. Can you make this clear?

The sentence was removed as it was redundant and could cause confusion. All relevant model descriptions are already included in the text.

L69: "factors such as" can be deleted here, right? Because the list is exhaustive?

Agreed.

L84: "This allows the computation of OA particle capacity to serve as a cloud condensation nuclei" This seems to suggest that OA particles are externally mixed with the rest of the particle population, which should not be the case?

OA are indeed internally mixed with the rest of the aerosol components within each size mode. In total, EMAC considers 7 particle populations described by lognormal size modes (four hydrophilic and three hydrophobic). The aerosol composition within each mode is uniform in size (internally mixed) but can vary between modes (externally mixed).

The way we account for the effect of OA on cloud droplet activation is by allowing it to affect the effective aerosol hygroscopicity of each size mode (here the 4 hydrophilic modes). Therefore, by explicitly simulating the chemical conversion of organic gases from initial emissions to a highly oxygenated state during photochemical ageing, ORACLE facilitates the tracking of changes in OA hygroscopicity resulting from these reactions, which in turn affect the CCN activity of the particles.

L86: Only in tropical regions? Are all the studies referenced below from tropical regions? What is the reference for the tropical region study? Consider revising these two sentences to make it clearer.

The text was reformulated for clarity. This was a simple evaluation of ORACLE's performance in simulating OA concentrations, using the same setup as in this study. The data was sourced from the tropics, as reported in the cited papers.

L94: Strike "however," as there is no contradiction here.

Amended.

Section 2.1: This section needs information here about new particle formation (NPF) and what role organics play there as well as SOA mass yields from BVOCs.

This information was added in the text as mentioned above.

L130: Out of interest, what vegetation would this result in? Is there grass land and crops in the region?

LPJ-GUESS simulates 12 plant functional types (PFTs). The implemented deforestation routine eliminates tree PFTs in regions identified by the HYDE land use data. In fully deforested grid cells, trees will never establish, as they are removed after each simulated year. In these areas, only C3 and C4 grasses, representing grassland/crop, can grow. We have clarified the crop-to-grass approximation and noted that, in the current setup, shrubs, which may be present in some grasslands, are not simulated.

L136: Which "fluxes" are referred to here?

This refers to $CO_2$ canopy fluxes. Text amended.

140    L137: I think this point needs to be highlighted more (as mentioned above as well). Where does the albedo and the roughness come from? What model is used for these properties? Are the land properties which interact with the meteorology from present day land use or something else? One thing is that the climate effects via albedo, roughness etc. are not included, another is that there might be feedbacks also on the BVOC emissions. I think this deserves more space both here and in the discussion.

145    This is now clarified in the text."The roughness length and albedo are kept as constant background values. Albedo is derived from satellite climatologies, while the roughness length is based on subgrid-scale orography and satellite-derived vegetation climatology. Vegetation changes do not provide feedback to the hydrological cycle. We use the native bucket model in ECHAM5, which employs fixed climatological vegetation (Hagemann, 2002). In this setup, BVOCs are interactive tracers that can be oxidized to
150    form secondary organic aerosols. This means that BVOCs can influence the oxidant chemistry of the atmosphere; however, we do not quantify such impacts in this work, focusing solely on aerosol changes."

   L174: I am not sure what "surface offline emissions" means. Is it aerosol and aerosol precursors? And is it then all except the BVOC emissions? Also, please give a reference to input for reproducibility.

155    A detailed description of the emissions is now included.

   L182: strictly speaking c and d do not show this.

   Agreed. Sentence is more clear now. "Fig. 2 shows the PNV scenario's spatial distribution of the vegetation fraction for tree (a) and grass (b) PFTs."

   L185: As far as I understand, grass is here used to represent both grass land and crops. Could you
160    comment on how realistic this is in terms of BVOC emissions?

   We included few sentences to explain this limitation. We also mention that LPJ-GUESS does not simulate shrubs which may influence the BVOC emission rates in natural grass land.

   Fig. 2: "Area of vegetation per unit ground area": is this the area covered by vegetation? Not e.g. leaf area index?

165    It is the area covered by vegetation. Text updated for clarity.

   Fig.3: Are the units correct in e and f, or is this also supposed to be Tg(C) as in the above plots?

The unit are correct. Panels a-d showing the BVOC emission flux in milligram per square meter per day. Panels e and f are global monthly means in teragrams. Note that we consider the total mass of BVOCs, not only the carbon mass.

170 Fig. 4: 5th-95th percentile of what? The annual and spatial mean? The monthly and spatial mean?

Updated to: "5th–95th percentile of the spatial mean over the 10 years simulated"

Fig. 5: A bit the same question: the spatio-temporal mean is just one number, it doesn't have a standard deviation? Maybe I misunderstood something? Also, the font size is a bit on the tiny end here.

The grey lines represent the relative difference between the two simulations ((exp1-exp2)/exp2). This
175 yields a 2D array (spatial dimension) of relative differences for each level. This array is then averaged across the spatial dimension, with the standard deviation used for the grey shading. A bug was identified in the computation of the standard deviation, and the figures have been updated accordingly. However, in Fig. 6, the standard deviation shading for the vertical profiles of CDNC was excluded because it was excessively large and completely overwhelmed the scale. This has now been addressed in the text.

180 L210: If you want, I would think you can remove the explanations of AOT and aerosol extinction since it should be text book material. But it's fine either way.

Agreed and removed.

L216: The sentence about which optical properties EMAC simulates seems out of place: consider deleting?

185 This sentence is quite important as it highlights the aerosol components that are used in EMAC when we refer to the *total* aerosol AOT.

L215-220: An analysis of the significance of these differences would be appropriate here.

Statistical significance analysis is now included.

Fig. 6: the relative plots look a bit strange in the bottom here. Why is there no variability in f?

190 This figure was revised. Standard deviation shading in the vertical profiles for CDNC was excluded.

L225-231: Again, it would be good to get some idea of the significance of these results, in particular the tiny increase in CCN and CDNC in high latitudes.

This is now discussed in the main text.

Fig. 7: Again, it would be useful to shade non-significant changes here.

Included.

L234: Check sentence structure.

Rephrased. "We estimate a global increase in aerosol radiative effect ($RE_{ari}$) of 60.4 mW m$^{-2}$, with the tropics showing a notably higher increase, averaging 91.2 mW m$^{-2}$."

L235: Figure 7 is not referenced here? Also, it feels like the blue blob over south-eastern Asia should be mentioned here already?

Fig. 7 is now referenced in the text. We also mention the cooling effect over SE Asia emerging from the increased aerosol extinction in this region.

L267-268: Insignificance should be calculated. Also, I assume the percentage change is calculated from (Fclean – Fclean,clear-sky) in both simulations. This is the total cloud radiative effect for each simulation, not the total effect of aerosols on clouds, and in my opinion using relative change doesn't make sense for this (just like it doesn't make sense for temperature).

Statistical significance is now incorporated and discussed in the text. Additionally, the sentence discussing radiative effects has been streamlined for consistency with the terms REari and REaci. The percentage change was eliminated in the text but kept in Table 1 for completion.

L279: Does the study confirm a decline in tree cover? It seems to me like the model is forced to have a reduction in tree cover, so this seems to be not a result, but an assumption in the study.

Good point. It is now emphasised that this was an assumption in the experiment, but can be validated by comparing values simulated by LPJ-GUESS with literature values.

L284: It says "studies" but there is only one study referenced. Also, I assume you mean global BVOC emissions?

More studies have been referenced.

L294: Why is this particularly true because the study only looks at land use changes? I think it would be good to say something here about which studies include temperature feedbacks and which are purely forcing for example, and in which direction this would contribute (lower or higher change).

220 The key point here is that PNV versus present-day vegetation differs from preindustrial vegetation versus present-day vegetation. Our simulations account for vegetation changes not only from preindustrial times but also from earlier periods. Although cropland and grazing land expansion was not as significant as during or after the industrial era, it did exist on a smaller scale. This clarification has been included.

The role of temperature feedbacks on BVOC emissions is also discussed here.

225 L294: MEGAN? It says MEAGN.

MEGAN, thanks.

L325: I don't quite follow the explanation here: Is it the growth of particles into the Greenfield gap and thus expanded life time that drives this, or the growth of particles into the size range where they have high light scattering efficiency?

230 The referee is correct. The growth of particles into the Greenfield gap reduces their deposition velocities and scavenging, extending their atmospheric lifetime. However, the main driver of increased aerosol extinction is their growth into the size range with higher light scattering efficiency, especially in the accumulation mode. While both factors contribute, enhanced scattering is the dominant influence on aerosol optical thickness.

[Figure]

235 The image above illustrates changes in aerosol optical thickness (AOT) for individual aerosol species. As expected, organic carbon (OC) extinction decreases across the board, particularly in tropical regions,

reflecting the reduced BVOC emissions in DCGL compared to PNV. Despite the overall decline in AOT attributed to OC, the AOT from WASO and H2O increases in some regions, notably Southeast Asia. The increase in total aerosol AOT, shown in Fig. 5e, therefore results from the rise in WASO and H2O AOT.

240

GMXe features 7 log-normal modes, with particles able to move between these modes. By examining aerosol mass and number across the modes in this region, we observed an increase in accumulation mode (mass) and a decrease in Aitken mode, indicating particle growth and a shift in the aerosol size distribution, resulting in greater extinction.

245 I also don't particularly follow the argument that the absence of organics drives the growth. It would be helpful to know something about how NPF and early particle growth was treated here. It would also be good to see some evidence for these explanations – are you sure about them? In general, I suggest rewriting this paragraph to make the argument clearer.

The presence of organics would provide more surface area for condensation, promoting growth in the
250 Aitken mode. In their absence, condensation is enhanced on the available accumulation mode particles, causing these particles to grow larger.

I also wonder why this effect does not show up when you compare the DCL and DCGL runs.

The fact that we don't observe this effect in the afforestation scenario supports our hypothesis. In this case, the increase in organics leads to more condensation in the Aitken mode, preventing sufficient
255 growth in the accumulation mode to shift the aerosol size distribution.

L335: "Changes in cloud properties . . . " Is this sentence needed?

Removed.

L336: Note that Scott et al (2014) look at surface level CCN and CDNC at cloud height, so it's not really comparable to column burden. Consider revising.

260 It was clarified that Scott et al (2014) looked at surface CCN and CDNC at approx. 900 hPa.

L338: I guess one reason why the change in CCN is so much larger than the change in CDNC in your study is that you are looking at the change in column burden CCN, and a lot of these particles might be too high up to interact with clouds?

Further analysis confirms this observation. The image below features diagonal crosshatching to indicate
265 areas that are not statistically significant, based on a two-tailed Student's t-test with a 90% confidence

level. Instead of considering the column burden of CDNC, this analysis focuses on the pressure level at which the difference in CDNC between the two simulations is maximized, averaging around 700 hPa, which is approximately 3.3 km. As shown in the image below, the CDNC changes at this altitude are statistically significant.

[Figure]

270 L241: Why do these results align well with Scott et al (2018)? Scott et al (2014) was the paper discussed previously in this paragraph, so should it still be Scott et al (2014)? If so, the question stands: I don't think the results are quite comparable, so I am not sure you have reason to say either that it aligns or does not currently?

Agreed. Originally I meant Scott et al (2014), but we agree that the results are not really comparable.
275 This sentence was removed.

L345: This is in addition to roughness changes and albedo changes that can also influence cloud cover (see e.g. discussion in Cerasoli et al. (2021, PNAS)).

Included in text.

L350: Is Tilmes et al. (2019) the best reference for such a general statement? It's a one model study.
280 Maybe a review paper would be better.

More studies have been referenced.

L353-356: Consider revising. It's a bit unclear what comes from which study and what are their assumptions.

Text revised accordingly.

285 L355: I would suggest sticking with ARI here instead of using direct and indirect effect.

Updated. I am now using only ARI and ACI for consistency.

L359: This would be a good place to discuss the impact of SOA yields (see e.g. discussion in Sporre et al. 2020, ACP) and potential oxidant chemistry (see e.g. Weber et al., 2022, Nat. Comms.).

SOA yield sensitivities are discussed in more detail in the discussion.

290 "The comparatively small ARI effect reported here from deforestation relative to the PNV scenario (60.4 mW m$^{-2}$) may stem from differences in methodology compared to other studies, which focus on the effects of full deforestation or total SOA (not bSOA). However, our RE$_{ari}$ estimates might be underestimating the actual impact due to comparatively lower bSOA yields in our model runs. The radiative forcing from BVOCs is strongly influenced by SOA yields, which can vary considerably with environmen-
295 tal conditions. SOA yields are sensitive to factors such as temperature and oxidant concentrations, which can alter the overall aerosol burden and their radiative properties (Sporre et al., 2020). Furthermore, the role of oxidant chemistry is crucial in determining the oxidation pathways and the formation of SOA. Variations in oxidant levels, particularly ozone and hydroxyl radicals, can substantially affect SOA formation rates and chemical composition (Weber et al., 2022)."

300 L366-377: Why is this not part of the manuscript? These are results, not discussion. Also, it is not clear what has gone into the Student' t test (Is a paired student t test with annual global means e.g.?). Furthermore, statistical significance of a result is not a property of the effect (large, small etc.), but only a measure of if it can be detected or not (how likely is the result to be by accident). I write this because the text as it stands seems to suggest that bSOA either is or is not significantly affecting
305 clouds, but this is highly dependent on model setup and if you run enough years, even a very small effect can be significant. Perhaps I am just misunderstanding the text, but please consider revising this paragraph.

Thanks for this comment. Statistical significance is now discussed in the results section following the updated figures including hatches. We use a two-tailed Student's t-test with 90% confidence ($p < 0.1$)
310 on annual means to minimise noise from sub-annual variability. This paragraph was updated accordingly.

L387: Is WASO changing its meaning? It's defined as water-soluble organic compounds earlier.

Typo, thank you.

**References**

315 Hagemann, S.: An improved land surface parameter dataset for global and regional climate models, 2002.

Sporre, M. K., Blichner, S. M., Schrödner, R., Karset, I. H., Berntsen, T. K., Van Noije, T., Bergman, T., O'donnell, D., and Makkonen, R.: Large difference in aerosol radiative effects from BVOC-SOA treatment in three Earth system models, Atmospheric Chemistry and Physics, 20, 8953–8973, 2020.

Tsimpidi, A., Karydis, V., Pozzer, A., Pandis, S., and Lelieveld, J.: ORACLE (v1. 0): module to simulate the organic aerosol
320 composition and evolution in the atmosphere, Geoscientific Model Development, 7, 3153–3172, 2014.

Weber, J., Archer-Nicholls, S., Abraham, N. L., Shin, Y. M., Griffiths, P., Grosvenor, D. P., Scott, C. E., and Archibald, A. T.: Chemistry-driven changes strongly influence climate forcing from vegetation emissions, Nature Communications, 13, 7202, 2022.

---

## Author Comment (AC2)

Author's response to comments from Anonymous Referee #2:

**"Land use change influence on atmospheric organic gases, aerosols, and radiative effects"**

by Ryan Vella et al.

5   We thank editor and referees for taking the time to review our manuscript and for the valuable feedback. Here, the comments from Anonymous Referee #2 (from August 07, 2024) are reproduced in black, while our comments are presented in blue.

**From Anonymous Referee #2's response:**

The aim of the manuscript is to detail changes in the impacts of biogenic emissions under different levels
10  of deforestation. The global study changes the area of land currently defined as tree plant functional types towards more grass and cropland area as a future world requires more land for agriculture. The authors find that the reduction in biogenic emissions from deforestation lead to associated reductions in secondary organic aerosol production which leads to warming. In an alternative scenario, the impacts of an extreme reforestation scenario are explored where a cooling effect is produced from increased biogenic
15  emissions.

I thank the authors for embedding the figures in the text, this greatly helps the reviewer!

The paper is well within the scope of ACP and I recommend publication after considering a few points.

    Thank you for your comments and recommendation for publication. Your concerns are addressed in
20   detail below.

1. Understanding the scenarios

I had real problems understanding the scenarios. I first thought PNV was the current vegetation. An explanation of what potential natural vegetation is would help when it is introduced on line 127 (i.e. the absence of humans).

25   PNV is now defined in the text. "PNV refers to the type of vegetation that would naturally occur in a specific area under certain climate, soil, and environmental conditions without human influence."

In the methods section it says DCGL is sometimes referred to as 'present day deforestation' on line 163. Yet on line 241 it says 'present-day land cover'. Perhaps this should be called 'current land cover'? It took me a couple of reads to work this out.

30   We agree that the term "present-day deforestation" might be misleading. We now only use "present-day land cover" throughout the manuscript.

I guess what I'm trying to get at is the study where we compare the current conditions the world finds itself in with either deforestation of afforestation, as it shows us what differences can be made if we collectively choose to adopt either scenario now. I'm finding the study using PNV confusing as it never
35   existed – certainly not in the years 2000-2012.

We acknowledge the potential confusion stemming from comparisons with different baselines. In the first comparison, PNV serves as the baseline against present-day conditions, while in the second, present-day serves as the baseline against reforested grazing land. The aim of this study is to highlight how current land cover influences BVOC emissions and to assess how an extreme reforestation scenario would perturb
40   these emissions and atmospheric states. We opted to nudge the meteorology over a "random" timeframe (2000-2012) to suppress feedbacks that could complicate the disentangling of climate responses purely resulting from land cover changes. Text has been added in Section 2.3 to address these concerns.

2. Other comments

The responses of PNV and DCL look very similar to me – in figures 3 (e,f) and figure 8 (e,f). Is this
45   because crops are not large bVOC emitters?

The annual cycle of isoprene and monoterpene emissions from PNV and DCL are very similar, differing only in magnitude, particularly in spring and summer (more emissions in PNV compared to DCL). This occurs primarily for two reasons: (1) the perturbation from cropland expansion is relatively small compared to that from grazing land expansion (see Fig. 1), and (2) the impact on BVOC emissions
50   depends significantly on the type of vegetation (PFT) being deforested and its associated emission rates. For instance, the conversion of natural grasslands to cropland would affect BVOC emissions differently than the conversion of tropical rainforests.

And why do the isoprene responses in both figures peak in spring as opposed to summer months?

MEGAN includes a leaf age factor, which accounts for reduced emissions for young and old leaves based
55   on observed LAI change. This explains the slight decrease in MEGAN emissions from April to May to June. This is discussed in more detail in Vella et al. (2023).

Line 166. The years being simulated should be stated at the start of the paragraph (rather than at line 170). And is the same meteorology driving all 3 of your scenarios, or is the model being driven by data

from a climate run for an assumed future extreme afforestation run? Some comment on the associated
60  impacts of warming on biogenic emissions and bSOA production is warranted.

The first sentence of the paragraph now specifies the years simulated (2000-2011). Text now highlights that we employ the same climate states in all scenarios, only perturbing land cover.

Figure 1. The unit is 'land transformation faction'. Does this figure already include land that is agricultural, or is this the land that is being changed in the model to crops and grazing land? I was surprised
65  given the large land transformation fraction (what I understand as 'deforestation' from the figure caption) in the mid-USA that we don't see more of an impact there in figures 5 onwards?

Figure 1 shows the land being converted from natural vegetation (PNV) to cropland or grazing land, primarily by removing tree PFTs in these regions. You are correct; this refers to areas of deforestation.

70  The HYDE dataset highlights significant deforestation signals in the central and northern USA. However, the impact on vegetation changes in the central USA is much weaker compared to the land cover fraction maps. This is because the region is naturally dominated by temperate grasslands, so the effect of deforestation is minimal here (since we are only removing tree PFTs). The more noticeable signal comes from deforestation further north, where temperate forests are being cleared.

75  The strongest signals in BVOC emissions and aerosol burden are observed in the tropics due to higher BVOC emission rates from tropical forests, which are more productive and diverse compared to temperate or grassland ecosystems. Extensive deforestation in these regions leads to significant reductions in BVOC emissions, which, coupled with the tropical climate's sensitivity to these compounds, greatly impacts aerosol formation. The stronger perturbation in the tropics, coupled with the atmospheric conditions in
80  these regions, results in a comparatively weaker signal at higher latitudes. This same pattern, seen in the USA, also occurs in Europe.

Figures 5 & 6. The line diagrams are missing a legend to distinguish between the colored lines.

The legend for the vertical profile plots is positioned between panels (a) and (e). The figures have been revised to feature larger text, including an enlarged legend for improved visibility.

85  Line 281. 'by'

This section was revised in response to comments from reviewer 1.

**References**

Vella, R., Forrest, M., Lelieveld, J., and Tost, H.: Isoprene and monoterpene simulations using the chemistry-climate model EMAC (v2.55) with interactive vegetation from LPJ-GUESS (v4.0), Geoscientific Model Development, 16, 885–906, 2023.

---

## Editor Decision (ED1)

Abstract line 6:

Suggestion to add "model" after chemistry-climate to improve clarity ("…a global atmospheric chemistry-climate model coupled with a dynamic global vegetation mode…"

Lines 43-45:

The processes being described are not necessarily unique to biogenic precursors. Minor wording changes are suggested to make that, and the rest of the paragraph, clearer:

As a result, monoterpene precursors may have distinct climate impacts given their influence on the aerosol numbers. Through condensational growth, bSOA participates in the absorption and scattering of solar short-wave radiation, contributing to aerosol-radiation interactions (ARI). Furthermore, newly formed bSOA particles can grow into sufficient sizes to activate as cloud droplets, thereby modifying cloud properties such as albedo and lifetime, and effectively contributing to aerosol-cloud interactions (ACI) (Forster et al., 2007).

Question: One of the reviewers asked about the role of biogenic compounds in NPF. As I understand it (based on your response and edits to the manuscript), in the GMXe model, biogenic compounds are not considered and do not affect number concentrations. They can only contribute to condensational growth. If this is correct, I suggest to add a sentence in the introduction (after lines 43-45), clarifying that only the contribution to ARI is considered in this work (or alternatively, that the contribution to REaci by this mechanisms is not considered). I do see that you report on changes in REaci. I'm not sure what the mechanism of change would be (since it is not enhanced NPF by biogenic precursors), and maybe that needs to be clarified in the results if it has not been already (e.g., in paragraph starting on line 273, 295, 461).

Line 300:

Suggestion to remove "Dot hatching…" sentence starting on line 300, since it was noted in the text already (for a different variable) and is noted in the figure captions.

---

## Author Response (AR2)

Author's response to comments from the editor:

**"Land cover change influence on atmospheric organic gases, aerosols, and radiative effects"**

by Ryan Vella et al.

The comments from the editor (from November 06, 2024) are reproduced in black, while our comments are presented in blue.

**From the editor's final response:**

The reviewers were unified in their evaluation of this manuscript, assigning scores of good to excellent in all of the primary review categories. The authors provided detailed responses to reviewer comments and made changes to the manuscript where appropriate. I have some minor questions and comments for the authors, largely focused on the areas of the manuscript where significant changes were made in response to the reviewers. Following minor revisions in response to those additional questions and comments, the manuscript to be suitable for publication.

We sincerely appreciate the insightful comments provided by the two anonymous referees, which we believe have significantly improved the quality of this work. We also thank the editor for their careful handling of the manuscript. Detailed responses are provided below.

P.S.: Throughout the manuscript, we used the term "land cover change (LCC)" rather than "Land use change". LCC is more widely used in the literature, and therefore I decided to update the title accordingly; from *"Land use change influence on atmospheric organic gases, aerosols, and radiative effects"* to *"Land cover change influence on atmospheric organic gases, aerosols, and radiative effects"*.

Abstract line 6: Suggestion to add "model" after chemistry-climate to improve clarity (" . . . a global atmospheric chemistry- climate model coupled with a dynamic global vegetation mode. . . "

Thank you for pointing out this typo. The text was updated to "...a global atmospheric chemistry-climate model..."

Lines 43-45: The processes being described are not necessarily unique to biogenic precursors. Minor wording changes are suggested to make that, and the rest of the paragraph, clearer: As a result, monoterpene precursors may have distinct climate impacts given their influence on the aerosol numbers. Through condensational growth, bSOA participates in the absorption and scattering of solar short-wave radiation,

contributing to aerosol-radiation interactions (ARI). Furthermore, newly formed bSOA particles can grow
into sufficient sizes to activate as cloud droplets, thereby modifying cloud properties such as albedo and
lifetime, and effectively contributing to aerosol-cloud interactions (ACI) (Forster et al., 2007).

Thank you. The text was updated to "As a result, monoterpene precursors may have distinct climate
impacts given their influence on the aerosol numbers. Through condensational growth, bSOA partici-
pates in the absorption and scattering of solar short-wave radiation, contributing to aerosol-radiation
interactions (ARI). Furthermore, newly formed bSOA particles can grow into sufficient sizes to acti-
vate as cloud droplets, thereby modifying cloud properties such as albedo and lifetime, and effectively
contributing to aerosol-cloud interactions (ACI) (Forster et al., 2007) In this study, we do not include
organic new particle formation (NPF) and focus only on the role of organic precursors in supporting
aerosol condensational growth."

Question: One of the reviewers asked about the role of biogenic compounds in NPF. As I understand
it (based on your response and edits to the manuscript), in the GMXe model, biogenic compounds are
not considered and do not affect number concentrations. They can only contribute to condensational
growth. If this is correct, I suggest to add a sentence in the introduction (after lines 43-45), clarifying
that only the contribution to ARI is considered in this work (or alternatively, that the contribution to
REaci by this mechanisms is not considered). I do see that you report on changes in REaci. I'm not sure
what the mechanism of change would be (since it is not enhanced NPF by biogenic precursors), and
maybe that needs to be clarified in the results if it has not been already (e.g., in paragraph starting on
line 273, 295, 461).

We agree that further clarification is needed. In the setup used here, we account for the condensational
growth of secondary organic aerosols (SOA), which, as you noted, directly influences aerosol-radiation
interactions (ARI). Additionally, this growth affects CCN and CDNC, as condensation can enable particles
to reach the critical radius required to act as CCN. To enhance clarity, we have added the following
paragraph: "Although this model setup excludes organic NPF, aerosol growth via the condensation of
organics can still impact CCN and CDNC by increasing particle size. This growth pushes existing particles
closer to the critical radius required for them to act as CCN and activate into cloud droplets, as described
by Köhler theory in the activation parameterisation used by the CLOUD submodel in EMAC."

In the Discussion section the following sentence was added to point out the relatively weak signals in
CDNC and REaci suggested here may be indeed due to the lack of organic NPF descriptions in the
model: "The lack of organic NPF descriptions in the model may also underestimate the real influence of
bSOA on CDNC and REaci reported in this work."

Line 300: Suggestion to remove "Dot hatching. . . " sentence starting on line 300, since it was noted in
the text already (for a different variable) and is noted in the figure captions.

Thank you, this sentence is now removed.

**References**

65  Forster, P., Ramaswamy, V., Artaxo, P., Berntsen, T., Betts, R., Fahey, D. W., Haywood, J., Lean, J., Lowe, D. C., Myhre, G., Nganga, J., Prinn, R., Raga, G., Schulz, M., and Van Dorland, R.: Changes in Atmospheric Constituents and in Radiative Forcing, in: Climate Change 2007: The Physical Science Basis. Contribution of Working Group I to the Fourth Assessment Report of the Intergovernmental Panel on Climate Change, edited by Solomon, S., Qin, D., Manning, M., Chen, Z., Marquis, M., Averyt, K. B., Tignor, M., and Miller, H. L., Cambridge University Press, Cambridge, United Kingdom and New York, NY, USA, 2007.